# The hybrid RAVE complex plays V-ATPase-dependent and -independent pathobiological roles in *Cryptococcus neoformans*

**Jin-Tae Choi[1], Yeseul Choi[1], Yujin Lee[1], Seung-Heon Lee[1], Seun Kang[2], Kyung-Tae Lee[2]\*, Yong-Sun Bahn**[1]\*

**1** Department of Biotechnology, College of Life Science and Biotechnology, Yonsei University, Seoul, Korea,
**2** Korea Zoonosis Research Institute, Jeonbuk National University, Jeonbuk, Republic of Korea

\* lee.kt@jbnu.ac.kr (K-TL); ysbahn@yonsei.ac.kr (Y-SB)

**Data Availability Statement:** Proteomics data for Rav1-mCherry strain are available via ProteomeXchange with identifier PXD042074. All

## Abstract

V-ATPase, which comprises 13–14 subunits, is essential for pH homeostasis in all eukaryotes, but its proper function requires a regulator to assemble its subunits. While RAVE (regulator of H$^+$-ATPase of vacuolar and endosomal membranes) and Rabconnectin-3 complexes assemble V-ATPase subunits in *Saccharomyces cerevisiae* and humans, respectively, the function of the RAVE complex in fungal pathogens remains largely unknown. In this study, we identified two RAVE complex components, Rav1 and Wdr1, in the fungal meningitis pathogen *Cryptococcus neoformans*, and analyzed their roles. Rav1 and Wdr1 are orthologous to yeast RAVE and human Rabconnectin-3 counterparts, respectively, forming the hybrid RAVE (hRAVE) complex. Deletion of *RAV1* caused severe defects in growth, cell cycle control, morphogenesis, sexual development, stress responses, and virulence factor production, while the deletion of *WDR1* resulted in similar but modest changes, suggesting that Rav1 and Wdr1 play central and accessary roles, respectively. Proteomics analysis confirmed that Wdr1 was one of the Rav1-interacting proteins. Although the hRAVE complex generally has V-ATPase-dependent functions, it also has some V-ATPase-independent roles, suggesting a unique role beyond conventional intracellular pH regulation in *C. neoformans*. The hRAVE complex played a critical role in the pathogenicity of *C. neoformans*, and *RAV1* deletion attenuated virulence and impaired blood-brain barrier crossing ability. This study provides comprehensive insights into the pathobiological roles of the fungal RAVE complex and suggests a novel therapeutic strategy for controlling cryptococcosis.

## Author summary

Although the assembly regulator of the V-ATPase is critical for its normal function, its pathobiological roles in fungal pathogens are largely unknown, with only yeast and human studies available for structural and functional analyses. In this study, we investigated the function of the RAVE complex in *C. neoformans*. We found that the cryptococcal RAVE complex had a hybrid format of yeast RAVE and human Rabconnectin-3

other relevant data are within the paper and its Supporting information files.

**Funding:** This work was supported by National Research Foundation of Korea funded by the Korean government (MSIT) (2021R1A2B5B03086596, 2021M3A9I4021434, 2018R1A5A1025077 to Y.-S.B.; 2022R1A4A3022401, and 2022R1C1C2003274 to K.-T.L.) and by the Yonsei Signature Research Cluster Program (2023-22-0012 to Y.-S.B.). This research was also partly supported by the Strategic Initiative for Microbiomes in Agriculture and Food funded by Ministry of Agriculture, Food and Rural Affairs (918012-4 to Y.-S.B.) and by International Joint Research Grant by Yonsei Graduate School. The funders had no role in study design, data collection and analysis, decision to publish, or preparation of the manuscript.

**Competing interests:** The authors have declared that no competing interests exist.

complexes and played diverse roles in growth, cell cycle control, morphogenesis, sexual reproduction, virulence factor production, stress responses, and virulence. Furthermore, we demonstrated that the hRAVE complex acts as an assembly regulator of the V-ATPase to maintain pH homeostasis in intra- and extracellular compartments and has unique roles in several stress responses in *C. neoformans*. These findings provide novel insights into the pathobiological functions of the RAVE complex and suggest its potential as a promising antifungal drug target.

## Introduction

Intracellular pH regulation and organelle acidification are essential in all eukaryotes to segregate the organelle's lumen from the cytoplasmic or extracellular space and to form a unique microenvironment for each organelle [1]. The representative examples include lysosomes and endosomes, which should maintain an acidic pH environment in their intracellular space separated from the extracellular space. Because all proteins depend on pH to maintain their structure and functions, intracellular pH homeostasis in these organelles is critical for diverse biological processes, such as receptor-ligand dissociation, hydrolase maturation, autophagy, and trafficking [2].

In all eukaryotes ranging from fungi to humans, vacuolar proton-translocating ATPase (V-ATPase) plays essential roles in regulating pH homeostasis and organelle acidification within intracellular and extracellular compartments [3]. In humans, the V-ATPases generate a membrane potential that drives neurotransmitter uptake and transport protons across the plasma membrane in some specialized cells like osteoclasts, renal intercalated cells, and epididymal clear cells [4–10]. Therefore, V-ATPase defects are associated with several diseases, such as tubule acidosis, osteopetrosis, and tumor metastasis [11]. The V-ATPases have an evolutionarily conserved structure and subunit composition in all eukaryotes. The V-ATPase consists of two domains: a $V_1$ domain as a cytoplasmic complex and a $V_0$ domain as a membrane-embedded complex [12]. The peripheral $V_1$ domain comprises eight subunits designated as A through H ($A_3B_3CDE_3FG_3H$) and is responsible for ATP hydrolysis. The integral $V_0$ domain consists of six different subunits a, d, e, c, c′, and c″ (yeast has an additional subunit c′) and functions in translocating protons from the cytoplasm to the lumen or extracellular space using the energy generated by the $V_1$ domain [13]. It has been reported that genetic and pharmacologic inactivation of V-ATPase changes intracellular pH and causes overall interference with various cellular processes, including protein processing and secretion [3,4].

In fungi, V-ATPases also contribute to cytosolic pH regulation [14]. In the opportunistic human fungal pathogen *Cryptococcus neoformans*, which is a primary etiological agent of fungal meningoencephalitis responsible for 180,000 deaths globally every year [14], the deletion of *VPH1* (the $V_0$ a subunit) results in reduced production of three virulence factors, including capsule, melanin, and urease. In support of this, the *vph1Δ* mutant is avirulent in a murine meningoencephalitis model [15]. In another opportunistic fungal pathogen, *Candida albicans*, the loss of *VMA11* (the $V_0$ c′ subunit) causes defects in iron acquisition required for survival within the host, and loss of *VMA7* (the $V_1$ F subunit) also results in defective *in vitro* filamentation and attenuates virulence during systemic candidiasis [16]. In the primary human fungal pathogen *Histoplasma capsulatum*, the insertional mutation of *VMA1* (the $V_1$ A subunit) abolishes virulence in a murine model of histoplasmosis [17].

The proper assembly of $V_0$ and $V_1$ and their regulation are critical for the normal function of the V-ATPase. Many extracellular stimuli and intracellular signals regulate the assembly

and disassembly of the V-ATPase. In response to environmental stimuli, including a high concentration of glucose or salts and pH changes, the assembly of V-ATPase is promoted in yeast. However, in contrast to yeast cells, glucose starvation induces an association with the V-ATPase in mammalian cells [18], suggesting that the regulatory mechanism of V-ATPase in yeast might differ from that of the mammalian species. Multiple intracellular signals and assembly factors, such as glycolysis, Ras/cAMP/PKA, cytosolic pH, PI(3,5)P$_2$, and RAVE complex, are intertwined with several extracellular stimuli and regulate the assembly and disassembly of V-ATPase [19].

Among the diverse regulating factors, the regulator of H$^+$-ATPase of vacuolar and endosomal membranes (RAVE) complex is a crucial regulator of V-ATPase assembly. In the budding yeast, the RAVE complex consists of three components; the adaptor protein Skp1 and its two subunits, Rav1 and Rav2 [20]. Rav1 and Rav2 are found only in the RAVE complex, whereas Skp1 can be associated with other cellular complexes. Rav1 constitutes the central component and binds to Rav2 and Skp1 [13]. Rav1 forms the interface between RAVE and V-ATPase subunits by using the WD40 repeat-containing domain, which is well known to be important in protein-protein interaction [21], serving as a starting point in regulating the assembly of V-ATPase [22]. In higher eukaryotes, the Rabconnectin-3 complex, a heterodimer composed of Rabconnectin-3α and Rabconnectin-3β subunits, performs the role of the yeast RAVE complex. The human Rabconnectin-3α proteins were named DMXL1 and DMXL2, which are orthologous to the yeast Rav1 protein. In contrast, the Rabconnectin-3β (also called WDR7) is not orthologous to the yeast Rav2 but can interact with different DMXL isoforms. The human Rabconnectin-3 does not have a Rav2-like subunit. In yeast, *rav2Δ* mutant cells display a phenotype similar to that of *rav1Δ* cells, indicating that Rav1 and Rav2 play functionally redundant roles [20,22]. Although yeast Skp1 is a highly conserved protein and a well-established subunit of the yeast RAVE complex [23], there is little evidence that Skp1 directly binds to Rabconnectin-3α or β, and mammalian Skp1 does not appear to be associated with pH regulation in organelles. Therefore, additional research on the components and roles of the RAVE/Rabconnectin-3 complex in eukaryotes is required.

In this study, we aimed to unravel the pathobiological roles of the RAVE complex in *C. neoformans*. Here, we discovered that *C. neoformans* has a hybrid form of the RAVE (hRAVE) complex, of which core (Rav1) and accessory (Wdr1) components are orthologous to yeast Rav1 and human Wdr7, respectively. Notably, the hRAVE complex plays V-ATPase-dependent and -independent roles in controlling growth and cell cycle, morphological changes, diverse stress response and adaptation, and virulence in *C. neoformans*. Therefore, our study provides new insights into the pathobiological roles of the fungal RAVE complex and suggests a possibility for developing novel antifungal drugs by targeting the RAVE complex that is more evolutionarily divergent than the V-ATPase itself.

## Results

### Identification of RAVE/Rabconnecntin-3 complex components in *C. neoformans*

The *S. cerevisiae* RAVE complex consists of Rav1, Rav2, and Skp1, with Rav1 being the central component [20,24]. Rav1 generally contains a single evolutionarily conserved Rav1 domain and multiple WD40 repeats. We identified a Rav1 ortholog in the *C. neoformans* genome database (CnRav1). Other fungi, including *C. albicans* (Ca), *Schizosaccharomyces pombe* (Sp), and *Ustilago maydis* (Um), also appear to have a single Rav1 ortholog, with size ranges from 1235 to 1518 amino acids (aa) (Fig 1A). The CnRav1 domain consists of 14 WD40 repeats, which span from the N-terminus to the central region, like other fungal Rav1 orthologs, except for

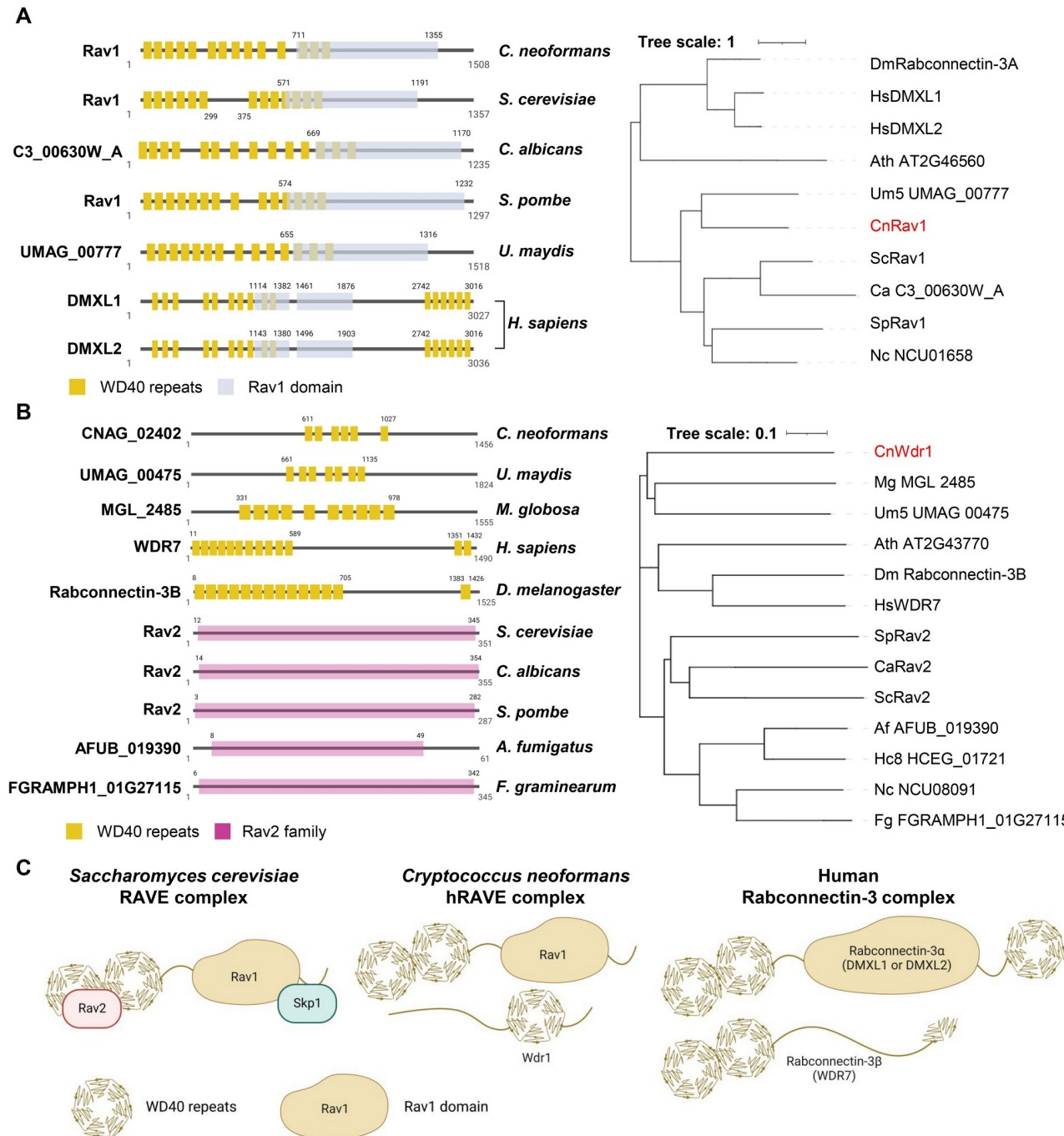

**Fig 1. The hRAVE complex is a hybrid of the yeast RAVE complex and the human Rabconnectin-3 complex.** The domain structure and phylogenetic tree of Rav1 and Wdr1/Rav2 orthologs in eukaryotes are illustrated in panels A and B, respectively, with abbreviations indicating the corresponding species: *U. maydis* (Um), *C. neoformans* (Cn), *C. albicans* (Ca), *S. cerevisiae* (Sc), *D. melanogaster* (Dm), *H. sapiens* (Hs), *A. thaliana* (Ath), *S. pombe* (Sp), *H. capsulatum* (Hc), *A. fumigatus* (Af), *M. globosa* (Mg), *N. crassa* (Nc), and *F. graminearum* (Fg). Domain information for Rav1 and Wdr1/Rav2 was retrieved from InterPro (https://www.ebi.ac.uk/interpro/), UniProt (https://www.uniprot.org/), and WDSPdb 2.0 (https://www.wdspdb.com/wdsp/). The domain IDs for each protein are provided as follows: Rav1 domain (IPR022033; RAVE complex protein Rav1 C-terminal), Rav2 family (IPR028241; RAVE subunit 2/Rogdi), and WD40 repeats (Uniprot KW-0853; WD40 repeat and WDSP-predicted WD40 repeat). (C) The expected model of the hRAVE complex in *C. neoformans* is shown.

ScRav1 (13 WD40 repeats). Human Rav1 orthologs (Raboconnectin-3α; DMXL1 and DMXL2) are structurally distinct from cryptococcal and other fungal Rav1 orthologs. DMXL1 and DMXL2 have a much larger size (3027 aa and 3036 aa) than fungal Rav1 orthologs and have 10 and 6 WD40 repeats at the N- and C-terminus, respectively, and two Rav1 domains in the central region. Several higher eukaryotes, including humans, have two Rabconnectin-3α subunit isoforms, but other eukaryotes, including *Drosophila melanogaster*, have only a single Rabconnectin-3α isoform [23]. Like structural differences, CnRav1 was phylogenetically closer to other fungal Rav1 orthologs than hsDMLX1 and hsDMXL2.

In contrast to Rav1, *C. neoformans* does not appear to have any yeast Rav2 ortholog. Instead, the fungal pathogen seems to have an ortholog (CNAG_02402) of Rabconnectin-3β (WDR7), which is structurally and phylogenetically distant from yeast Rav2 (Fig 1B). Interestingly, WDR7 orthologs are present in some fungal species, including *U. maydis*, *Mucor circinelloides*, and *C. neoformans*, but not in ascomycetous yeast species such as *S. cerevisiae*, *S. pombe*, and *C. albicans* (S1 Table). As a previous independent study named CNAG_02402 *WDR1* [25], we denoted it as *WDR1* instead of *WDR7*.

Another RAVE component Skp1 ortholog (CNAG_00829), was also present in *C. neoformans* (S1A Fig). However, Skp1 interacts with various protein complexes in addition to Rav1 of the RAVE complex [20,26,27] (S1B Fig). Supporting it, deletion of *SKP1* is not feasible in *S. cerevisiae*, suggesting that Skp1 is an essential yeast protein. Similarly, our multiple knockout attempts of *SKP1* in the haploid *C. neoformans* H99 strain failed to yield correct knockout mutants. Furthermore, we found that repression of *SKP1* expression by replacing its native promoter with the copper-repressible *CTR4* (copper transporter 4) promoter resulted in marked growth defects in *C. neoformans* (S1C and S1D Fig), implying that Skp1 could be a putative essential protein in the pathogen. To more definitively establish the essential role of Skp1 in the viability of *C. neoformans*, we generated heterozygous *SKP1/skp1Δ* mutants in AI187, a genetically engineered diploid strain [28] (S1E Fig). Despite examining over 50 spores through random spore analysis, we were unable to isolate any haploid spores with an *SKP1* knockout, thereby confirming the essential nature of Skp1 in *C. neoformans* (S1F Fig). Given these findings and existing reports suggesting that human Skp1 is unlikely to interact with DMXL1 and DMXL2 [23], we opted not to further investigate Skp1's role in functionally characterizing the RAVE complex.

All these data indicate that *C. neoformans* appears to have the hybrid form of the yeast RAVE complex and human Raboconnectin-3 complex because CnRav1 seems to be more orthologous to yeast Rav1 in terms of its protein size and structural configuration and CnWdr1, but not CnRav2, is present in *C. neoformans* (Fig 1C). In this sense, we named it the hybrid RAVE (hRAVE) complex because CnRav1 and CnWdr1 play major and minor roles in diverse pathobiological functions in *C. neoformans* and interact with each other, as described later.

## Roles of the RAVE complex in growth, cell cycle control, and morphogenesis

To elucidate the hRAVE complex's function, we employed a reverse genetics approach for CnRav1 and CnWdr1. To this end, we generated *rav1Δ*, *wdr1Δ*, and *wdr1Δ rav1Δ* mutants in *MATα C. neoformans* strain (H99) background through homologous recombination (S2 Fig). To verify their mutant phenotypes, we also generated *rav1Δ*::*RAV1* by reintegrating its wild-type allele into its native locus (S2 Fig). First, we compared the growth of *rav1Δ* and *wdr1Δ* mutants with that of the wild-type and complemented strains by spot assays at a varying range of temperature: 25°C, 30°C, 37°C, and 39°C. *RAV1* deletion led to a marked growth defect

even under nutrient-rich conditions [yeast extract-peptone-dextrose (YPD) medium] at 30°C, whereas *WDR1* deletion did not affect the growth (Fig 2A). At elevated temperatures (39°C), *rav1Δ* showed even more severe growth defects and even *wdr1Δ* exhibited an evident growth defect at 39°C (Fig 2A). Notably, the growth defect of the *rav1Δ wdr1Δ* double mutant was as defective as the *rav1Δ* single mutant (Fig 2A), suggesting that Rav1 and Wdr1 may play central and accessory roles, respectively, in the hRAVE complex. Quantitative growth rate measurement further confirmed its growth patterns (Fig 2B).

Apparent growth defects observed in *rav1Δ* strains implied that the hRAVE complex is likely involved in the cell cycle control of *C. neoformans*. Supporting this, fluorescence-activated cell sorting (FACS) analysis revealed that distinct cell cycle phases observed in the wild-type were absent in *rav1Δ* (Fig 2C). Unlike the wild-type cells, which had discrete n and 2n cell populations, the *rav1Δ* cells showed an aberrant ploidy distribution. In addition, the cell ratio of the $G_2$/ M phase was much lower in the *rav1Δ* mutant than in the wild-type strain (Fig 2D).

Defective growth and cell cycle control of the *rav1Δ* mutant led us to suspect that the hRAVE complex could be involved in the morphogenesis of *C. neoformans*. Therefore, we observed the cellular morphology of *rav1Δ* in nutrient-rich solid medium (YPD) and capsule-inducing solid media (Littman, DME, and RPMI). We briefly sonicated cultured cells and observed their morphology to discern whether such cell aggregation resulted from mere physical association among cells or cytokinesis defects. The *rav1Δ* cells were more aggregated than wild-type ones (upper panel in Fig 2E). Quantitative measurement confirmed this finding (lower panel in Fig 2E). In the wild-type, the vast majority of cells (>80%) only exist as single cells, and there were no four or more linked cells. In *wdr1Δ*, 40–70% of cells exist as single cells, and 5–20% of four or more connected cells exist. Yet, in *rav1Δ*, fewer than 20% of cells exist as single cells, and roughly 40–80% of four or more connected cells exist (Fig 2E). We also made a similar finding in broth media (S3A Fig). Interestingly, when cultured in Littman, DME, and RPMI media, *rav1Δ* cells showed pseudohyphal growth (upper panel in Fig 2E). In contrast to *rav1Δ*, most *wdr1Δ* cells exhibited similar cellular morphologies to the wild-type in the YPD medium, but some showed somewhat elongated cell morphology in the capsule-inducing media.

The RAM signaling pathway is highly conserved among eukaryotes and plays a role in diverse cellular processes, including cell cycle regulation, cell separation, mating, and cell polarized growth [29]. Unusual morphological changes of *rav1Δ*, which seem like pseudohyphal growth, allowed us to investigate a link with the RAM pathway related to pseudohyphal development in *C. neoformans* [30]. We compared *rav1Δ* with *cbk1Δ* and *kic1Δ* in the RAM pathway in several solid growth and capsule-inducing media. Both RAM pathway mutants showed pseudohyphal growth in all media, whereas *rav1Δ* showed pseudohyphal growth only in capsule-inducing media (Littman, DME, RPMI, FBS) and not in YPD and YNB media (S3B Fig). In addition, the pseudohyphal form of *rav1Δ* also differed from that of RAM pathway mutants: *cbk1Δ* and *kic1Δ* were long rod-shaped, but *rav1Δ* was more rounded and shorter. This indicates that Rav1 is involved in morphological changes, and its role appeared independent of the RAM pathway.

These results suggested that the hRAVE complex, in which Rav1 and Wdr1 are the central and accessory components, respectively, is required for the growth, cell cycle control, and morphogenesis of *C. neoformans*.

## Roles of the hRAVE complex in the sexual development of *C. neoformans*

The function of the hRAVE complex in the morphogenesis of *C. neoformans* prompted us to address its roles in yeast-to-filament morphological transition during the sexual differentiation

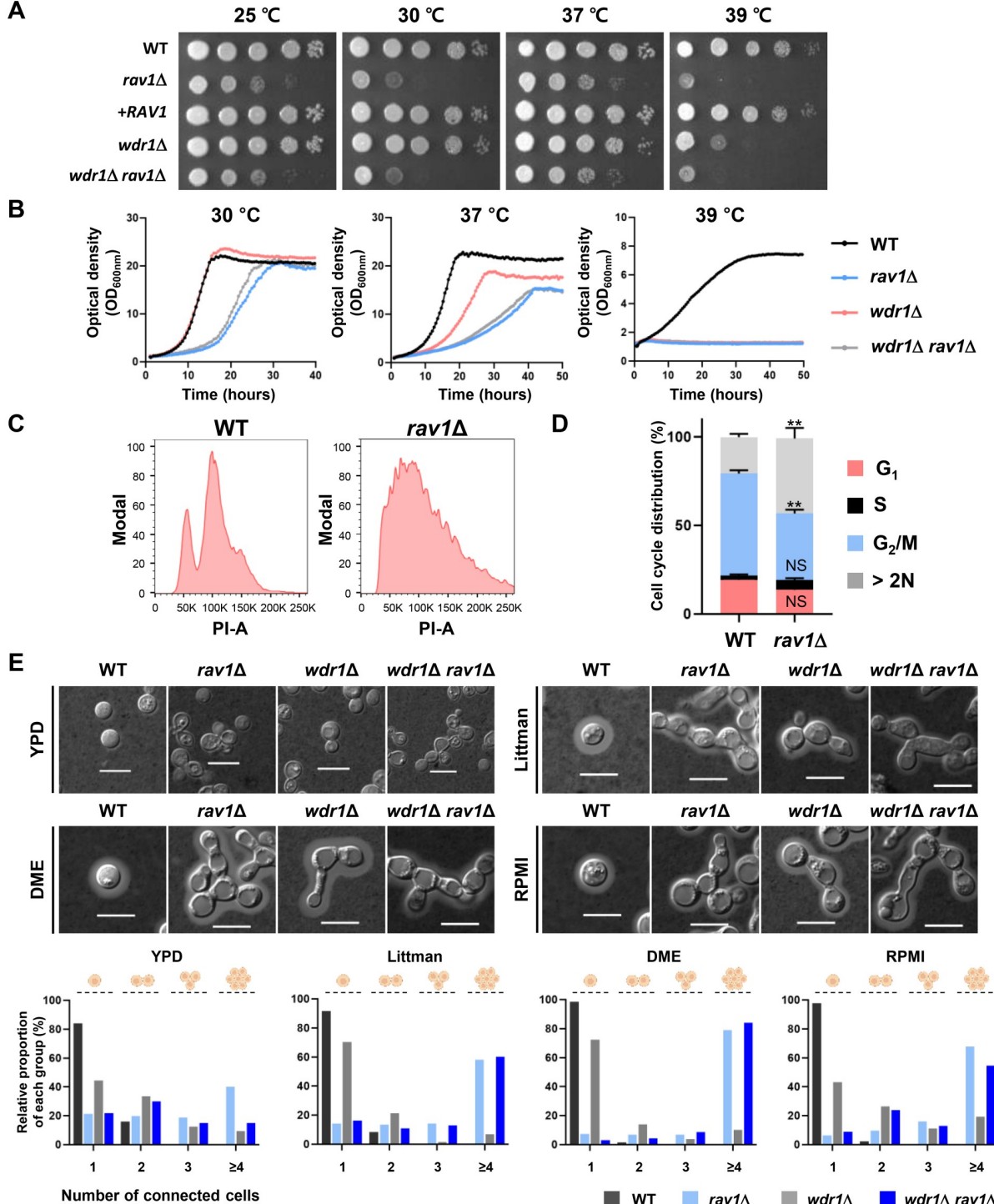

**Fig 2. The role of Rav1 in growth, cell cycle control, and morphological change.** (A) Qualitative spot assays showing growth and thermosensitivity of WT (H99), *rav1Δ* (YSB7589), *rav1Δ::RAV1* (YSB10754), *wdr1Δ* (YSB10032), *wdr1Δ rav1Δ* (YSB10604) strains. Plates were incubated for 2 days and photographed. (B) Quantitative growth rates of the same strains monitored at 30 ˚C, 37 ˚C, and 39 ˚C using a multi-channel bioreactor for 40 h. Each curve is a single representative of two independent experiments. (C, D) Flow cytometry analysis of cell cycle phases in propidium iodide (PI)-stained cells. Representative data from three biological replicates are shown. (D) The percentage of cell populations classified as interphase ($G_1$ peak and S phase) and mitotic phase ($G_2$/M peak) from the same experiment (C) was quantified and presented in the graph. One-way ANOVA with Bonferroni's multiple-comparison test determined statistically significant differences between the wild-type and mutants. Error bars indicate the standard error of the mean (SEM) (NS, non-significant; **, $P$ = 0.001–0.01). (E)

Quantification of unseparated cells measured for each *C. neoformans* strain cultured on YPD, Littman, DME, and RPMI agar media at 37˚C for 2 days. After picking a colony from the media, cells were washed twice with PBS, added with 0.5 ml PBS, and briefly sonicated. Capsules were stained with India ink, and the morphology of the wild-type and mutant strains were observed under a microscope (Bar, 10 μm). More than 100 cells were measured, and the graph shows the average of three biological replicates.

of the pathogen. To this end, we additionally generated *rav1Δ* and *wdr1Δ* mutants in the *MAT***a** strain (YL99) background (S4A Fig). In both unilateral (wild-type x mutant) and bilateral (mutant x mutant) mating with *rav1Δ*, filamentation was abolished entirely (Fig 3A). The fact that *rav1Δ* mutants did not show any filamentous growth even after 25 days of incubation on the V8 medium suggested that this result is not simply due to the growth defect of the *rav1Δ* strain. In support of this, the expression of *RAV1* increased about ten folds at 24 h post-mating, implying that *RAV1* could be involved in the mating process (Fig 3B). Moreover, the *wdr1Δ*, which had no growth defect at 25˚C, showed reduced filamentation in both unilateral and bilateral mating (S4B Fig), albeit weaker than the phenotype of *rav1Δ*, further strengthening the claim that the hRAVE complex could be involved in the mating process.

Next, we addressed which mating step the hRAVE complex is involved in. First, we measured the expression level of the *MFα1* pheromone gene during bilateral mating (Fig 3C). The *MFα1* expression increased about 15 folds at 24 h and 48 h postmating of *MATα* and *MAT***a** wild-type cells. In bilateral mating with *rav1Δ* mutants, however, *MFα1* expression did not

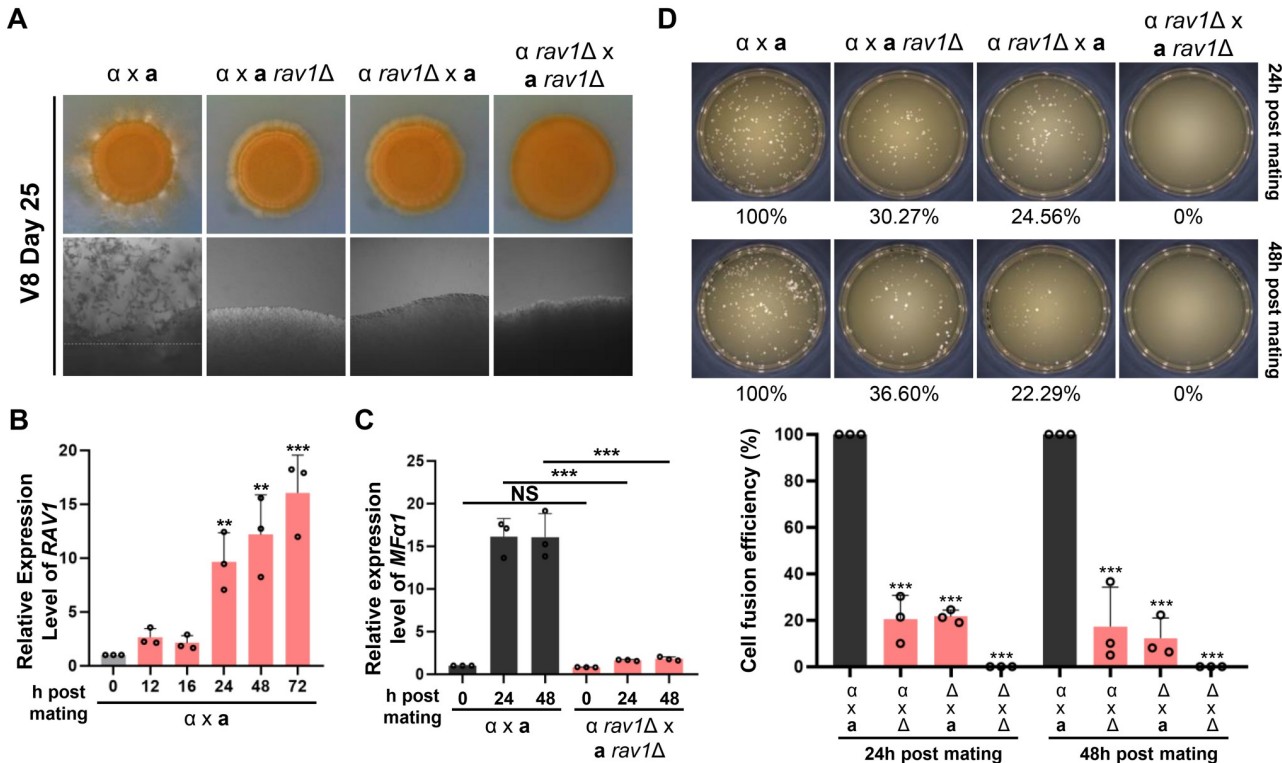

**Fig 3. The role of Rav1 in sexual differentiation.** (A) Qualitative photographs of cocultured *MATα* and *MAT***a** strains on V8 medium plates (pH 5.0) for 25 days at room temperature in the dark. Strains used for the mating assay are: α (H99) × **a** (YL99), α *rav1Δ* × **a** (YL99**a**), α (H99) × **a** *rav1Δ*, and α *rav1Δ* × **a** *rav1Δ*. (B) qRT-PCR analysis of the *RAV1* gene during the mating process. (C) qRT-PCR analysis of *MFα1* pheromone gene during the mating process. The expression level of each gene was normalized by that of *ACT1*. (D) Data from cell fusion assay. Control strains containing different selectable markers [*MATα* YSB119 (*NAT*) and *MAT***a** YSB121 (*NEO*)] were used. The plate picture is a representative of three biological replicates. Percentages normalized to the positive control are mean values from each experiment indicated by a dot. The error bars indicate the standard errors of the means (SEM). Statistical significance was determined using one-way ANOVA with Bonferroni's multiple-comparison test: ***, *P* < 0.0001.

significantly increase after 24 h and 48 h postmating (Fig 3C). Such defective pheromone induction during mating with *rav1Δ* mutants could affect cell-to-cell fusion. As expected, unilateral mating between the control and *rav1Δ* strain led to a significantly reduced cell fusion at 24 h and 48 h postmating. Bilateral mating with *rav1Δ* mutants led to a complete loss of cell fusion (Fig 3D). All these data suggest that the hRAVE complex is required to activate the early mating stage in *C. neoformans*.

## Roles of the hRAVE complex in cryptococcal virulence factor production

Next, we investigated whether the hRAVE complex is involved in producing two major virulence factors in *C. neoformans*: antiphagocytic polysaccharide capsule and antioxidant melanin pigment [31,32]. As shown in Fig 2E, the *rav1Δ* mutant appeared to produce wild-type capsule levels in most capsule-inducing media but slightly decreased in the Littman medium. However, quantitative capsule size measurement was not feasible due to the elongated, pseudohyphae-like morphology of the mutant. Therefore, we addressed whether Rav1 regulates the expression of capsule-related genes, such as *CAP10*, *CAP59*, *CAP60*, *CAP64*, *GAT201*, *YAP1*, *ADA2*, and *BZP4*, in Littman medium (S5A Fig). Although Rav1 did not significantly regulate the expression of most capsule-related genes, the expression of capsule-regulating transcription factor *GAT201* significantly increased under Littman-based capsule-inducing conditions in the *rav1Δ* mutant. As we recently reported that *GAT201* overexpression markedly enhances capsule production [33], *GAT201* expression appeared to be induced as a compensatory mechanism in the *rav1Δ* mutant.

In contrast to the minor role in capsule production, the hRAVE complex played a critical role in melanin production. The *rav1Δ* mutant showed even more severe defects in melanin production than the adenylyl cyclase mutant (*cac1Δ*) in all melanin-inducing media (Fig 4A). Quantitative measurement of laccase activity further confirmed this finding (Fig 4B). Moreover, melanin production of the *wdr1Δ* was also reduced compared to the wild type, indicating that the hRAVE complex is involved in melanin production (S5B Fig).

The *rav1Δ* mutant was as defective in melanin production as the mutant with deleted *LAC1*, which encodes a central melanin-biosynthetic laccase enzyme in *C. neoformans* [34]. We next addressed whether Rav1 is involved in the induction of *LAC1* and melanin-regulating transcription factors, *USV101*, *MBS1*, *HOB1*, and *BZP4*, under nutrient-starvation conditions (Fig 4C). In contrast to highly defective melanin production in *rav1Δ*, the expression of *LAC1*, *MBS1*, *HOB1*, and *BZP4* was overall increased under glucose-starved conditions in *rav1Δ* compared to wild-type (Fig 4C). All these data suggested that the hRAVE complex does not directly regulate melanin biosynthesis pathways and its inhibition may induce the melanin-regulating genes as a compensatory mechanism.

## Roles of the hRAVE complex in organelle acidification and vesicle trafficking

It is well known that the yeast RAVE complex is involved in organelle acidification and trafficking of the trans-Golgi network (TGN) and secretory/endocytic vesicles [13,22,24,35]. To address whether the hRAVE complex plays similar roles in *C. neoformans*, first, we monitored the cellular location of Rav1 by generating *rav1Δ::RAV1-mCherry* complemented strains, in which mCherry was in-frame fused to the C-terminus of Rav1. The *rav1Δ::RAV1-mCherry* strain was phenotypically identical to wild-type and *rav1Δ::RAV1* strains, indicating that Rav1-mCherry is functional (S6A and S6B Fig). When we stained the Rav1-mCherry strain with the LysoSensor dyes, which are acidotropic probes that accumulate in acidic organelles as the result of protonation, Rav1 appeared to localize to the cytosol and to be particularly

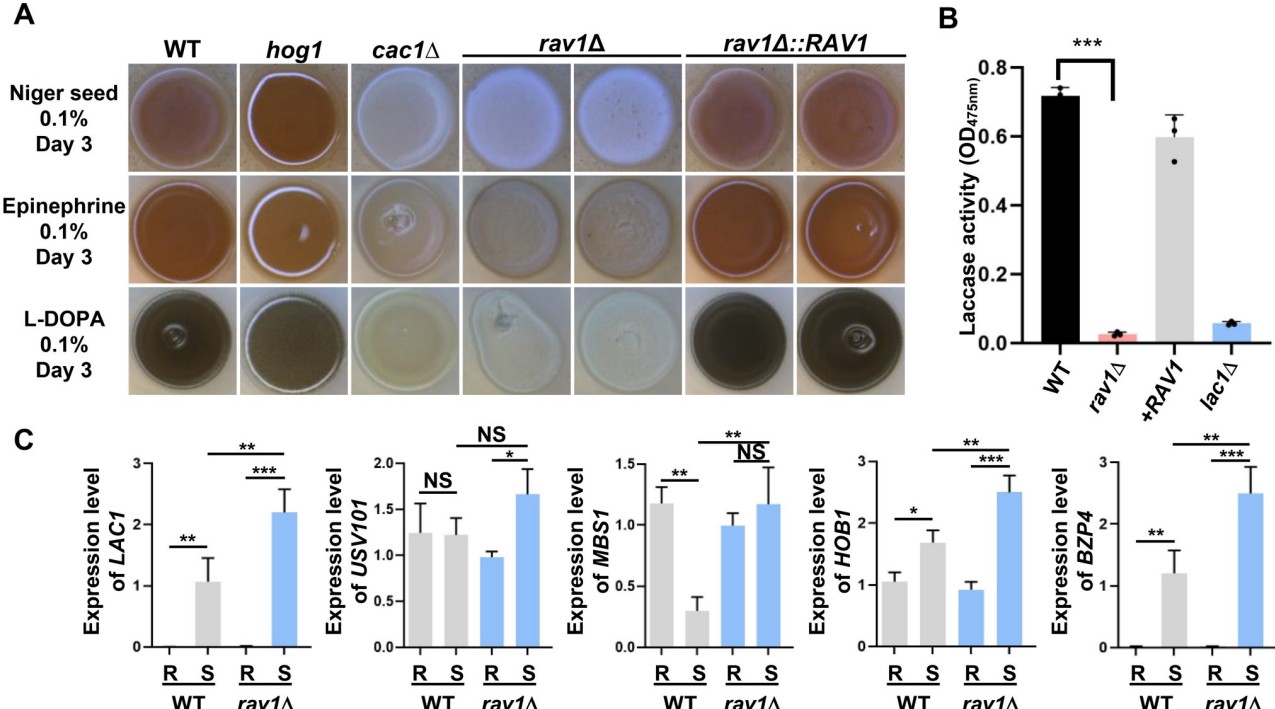

**Fig 4. The role of Rav1 in melanin production.** (A) Melanin production assay. The wild-type (WT; H99), *rav1*Δ (YSB7465 and YSB7589), *rav1*Δ::
*RAV1* (YSB10754 and YSB10755), *hog1*Δ (YSB64), and *cac1*Δ (YSB42) mutants were spotted onto Niger seed, epinephrine, or L-DOPA agar medium
containing 0.1% glucose, incubated at 37˚C for 3 days, and then photographed. Positive and negative controls were *hog1*Δ and *cac1*Δ strains,
respectively. Representative images from three independent experiments are shown here. (B) Laccase assays. WT (H99), *rav1*Δ (YSB7589), *rav1*Δ::*RAV1*
(YSB10754), and *lac1*Δ (CHM3) mutants were cultured in liquid YPD medium overnight, washed thrice with PBS, and incubated for 4 days in YNB
plus 0.05% glucose with 10 mM epinephrine. (C) qRT-PCR analysis of melanin production-related genes, *LAC1*, *USV101*, *MBS1*, *HOB1*, and *BZP4*,
using total RNA of WT and *rav1*Δ mutant strains under nutrient-rich (R; YPD) or nutrient-starved (S; YNB without glucose) conditions. For induction
of melanin production-related genes by nutrient starvation, cells were further incubated at 30˚C in YNB media for 2 h. One-way ANOVA with
Bonferroni's multiple-comparison test was used to determine the statistical significance of differences: *, $P < 0.05$; **, $P < 0.01$; ***, $P < 0.0001$; NS,
non-significant. Error bars indicate SEM.

enriched in the vacuole periphery as punctate forms (Fig 5A). Notably, such punctate signal
intensity of Rav1-mCherry around the vacuole membrane increased at high glucose concen-
tration, which agrees with increased acidification via high glucose levels like the case of *S. cere-
visiae* [36]. All these data indicate that *C. neoformans* undergoes organelle acidification upon
glucose addition, probably through the hRAVE complex.

The acidification of organelles is closely linked to protein sorting and organelle function
[36]. Ligand dissociation from their respective receptors takes place within a specific pH
range, which serves a crucial role in determining their final targeting [1]. Furthermore, the
luminal pH of endosomes can impact the association of trafficking factors [37]. For these rea-
sons, we monitored the impact of *RAV1* deletion in the cellular localization of the pheromone
transporter Ste6. In *S. cerevisiae*, the membrane localization of Ste6 is controlled by endosome
trafficking [38]. We also previously reported that cryptococcal Ste6 is localized to ER and
endosomes under vegetative growth conditions but highly enriched at the plasma membrane
upon switching to the mating condition [39]. Therefore, we deleted *RAV1* in the previously
constructed *STE6-GFP* strain (S7 Fig). We hypothesized that *RAV1* deletion could cause
defects in normal vesicle trafficking of Ste6 to the plasma membrane. Therefore, we cultured
*STE6-GFP* and *STE6-GFP rav1*Δ strains in YPD medium for vegetative growth and V8
medium for mating and stained them with the lipophilic styryl dye FM4-64 that binds to the

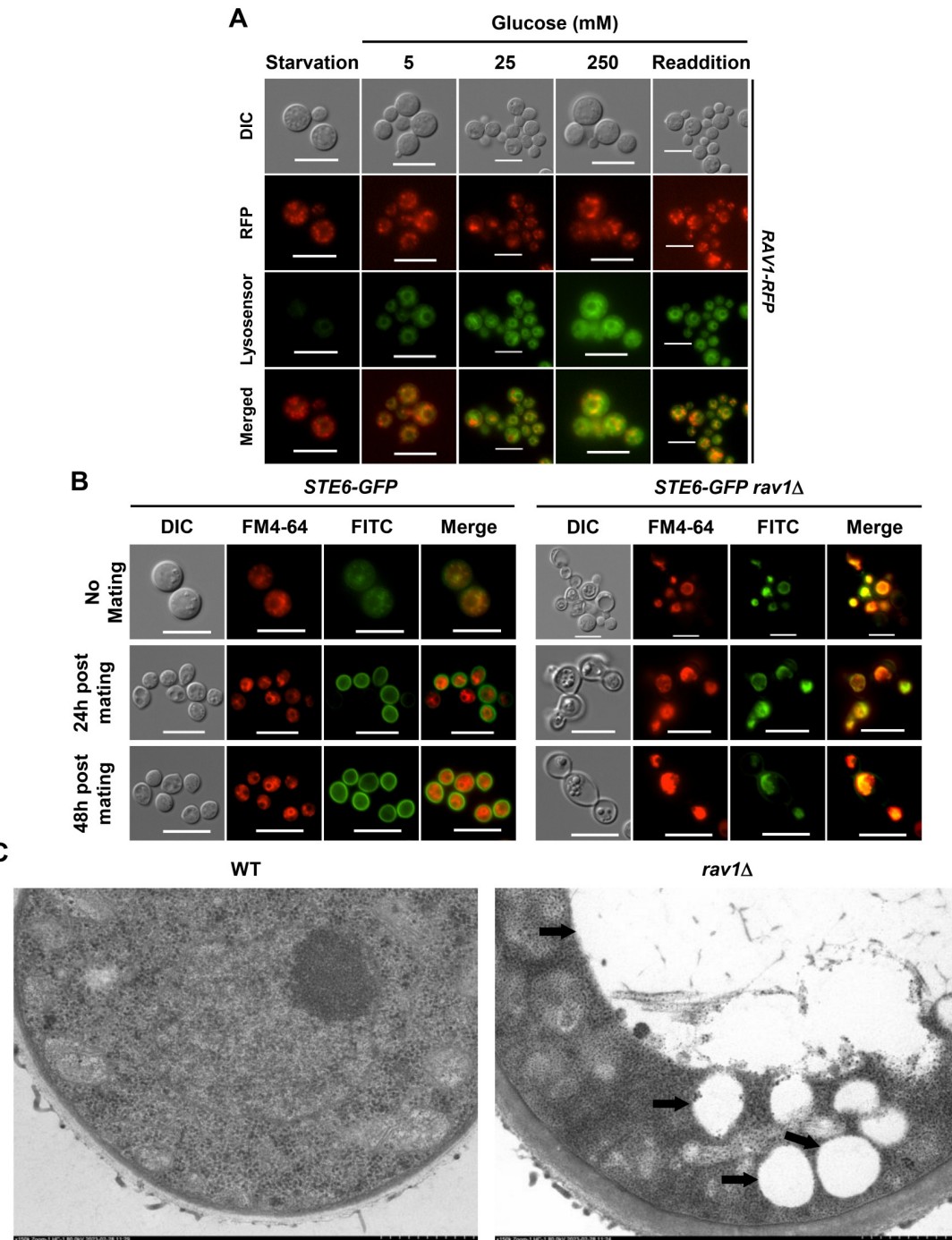

**Fig 5. The role of the hRAVE complex in vesicle trafficking.** (A) Acidification monitoring in Rav1-mCherry strain under different glucose concentrations. The strain was cultured in YPD overnight and then inoculated in liquid YNB medium without glucose (Starvation), with glucose concentrations of 5 mM, 25 mM, or 250 mM for one hour. Cells cultured without glucose (Starvation) were replaced with fresh YNB containing 250 mM glucose (Readdition) and cultured again for one hour. Acidification was measured using LysoSensor dyes as an acidotropic probe. Fluorescence microscopy was used to observe cells under each condition. One representative image from three biological replicates is shown here. Bars indicate 10 μm. (B) Localization of Ste6 in response to deletion of *RAV1* during mating. FM4-64 was used as a vacuole membrane-staining dye. All images are representative images of more than 50 cells observed and photographed. (C) Transmission electron microscopy (TEM) analysis of WT and *rav1Δ* mutant strain. The black arrow indicates the vacuole. Both images are shown at 150 k magnification.

plasma membrane and is then endocytosed and trafficked to the vacuolar membrane [40] (Fig 5B). As expected, after 24 h postmating, Ste6 localized to the cell membrane in *STE6-GFP* strains. However, in *STE6-GFP rav1Δ* strains, Ste6 appeared to be accumulated in the vacuole, not the cell membrane, even under mating conditions (Fig 5B). This phenomenon can explain why *RAV1* deletion abolished the mating of *C. neoformans* (Fig 3) because Ste6 is required for the extracellular release of pheromone that initiates the mating process.

To support these results, we performed transmission electron microscopy (TEM) for wild-type and *rav1Δ* cells. In comparison to the wild-type strain, the *rav1Δ* strain frequently demonstrated an accumulation of vesicles that increased the size and the number of vacuoles (Fig 5C). These observations support previous findings that the hRAVE complex regulates the vesicle trafficking of Ste6. Furthermore, defective melanin production in *rav1Δ* shown in Fig 4 also likely resulted from the perturbed vesicle trafficking of melanin-containing vesicles to the cell surface. All these data indicate that the hRAVE complex is essential for organelle acidification and vesicle trafficking in *C. neoformans*.

## Intracellular pH-dependent and -independent roles of the hRAVE complex

In yeast, deletion of any $V_0$ or $V_1$ subunits of V-ATPase leads to a very similar phenotype, called Vma$^-$ phenotype, with pH and calcium-sensitive growth, metal ion sensitivity, and inability to grow on non-fermentable carbon sources [3]. pH-dependent growth is one of the representative Vma$^-$ phenotypes, in which V-ATPase mutants are viable when grown in acidic media but cannot grow in alkaline conditions. It is also known that yeast RAVE complex mutants exhibit a partial Vma$^-$ phenotype, which is temperature-dependent and milder than in a V-ATPase mutant; yeast *rav1Δ* mutant shows growth defect at 37°C, but very slightly at 30°C, on pH 7.5-adjusted YPD medium [20,24]. However, we found that the *C. neoformans rav1Δ* mutant exhibited growth defects even at 30°C on pH 7.5-adjusted YPD medium but significantly restored growth on pH 5.5-adjusted YPD medium (Fig 6A and 6B). This result indicated that deletion of the hRAVE complex results in a more evident Vma$^-$ phenotype (e.g., pH-dependent growth) than the yeast RAVE complex.

Next, we further investigated the pH-dependent growth phenotypes of the hRAVE complex mutants under various stress conditions. For this purpose, we compared the growth of wild-type, *rav1Δ*, and *rav1Δ::RAV1* strains in YPD growth media having different stress-inducing agents under varying pH ranges (5.5, 6.8, and 7.5) (Fig 6C). Notably, we found that acidic pH adjustment of the growth medium differentially restored the resistance of *rav1Δ* to various stress conditions. First, acidic pH adjustment markedly restored the growth of *rav1Δ* in a YPD medium containing certain oxidative stress agents (*tert*-butyl hydroperoxide and menadione), genotoxic agents (methyl methanesulfonate and hydroxyurea), cell wall destabilizing agents (congo-red and calcofluor white), heavy metal stress agent (CdSO$_4$), or a polyene class of antifungal drug (amphotericin B) (Fig 6C). In contrast, acidic pH adjustment did not rescue or only weakly restored the growth defects in the YPD medium containing hydrogen peroxide (H$_2$O$_2$), thiol-specific oxidant (diamide), ER stress agents (tunicamycin and DTT), membrane-destabilizing agent (SDS), osmotic stress agent (sorbitol), or non-polyene classes of antifungal agents (fluconazole, fludioxonil, and flucytosine) (Fig 6C). All these data indicate that the hRAVE complex is likely involved in both pH-dependent and pH-independent phenotypes in *C. neoformans*.

## V-ATPase-dependent and -independent roles of the hRAVE complex

We aimed to discern whether the hRAVE complex operates through two distinct pathways: V-ATPase/pH-dependent and -independent mechanisms. Specifically, we focused on its

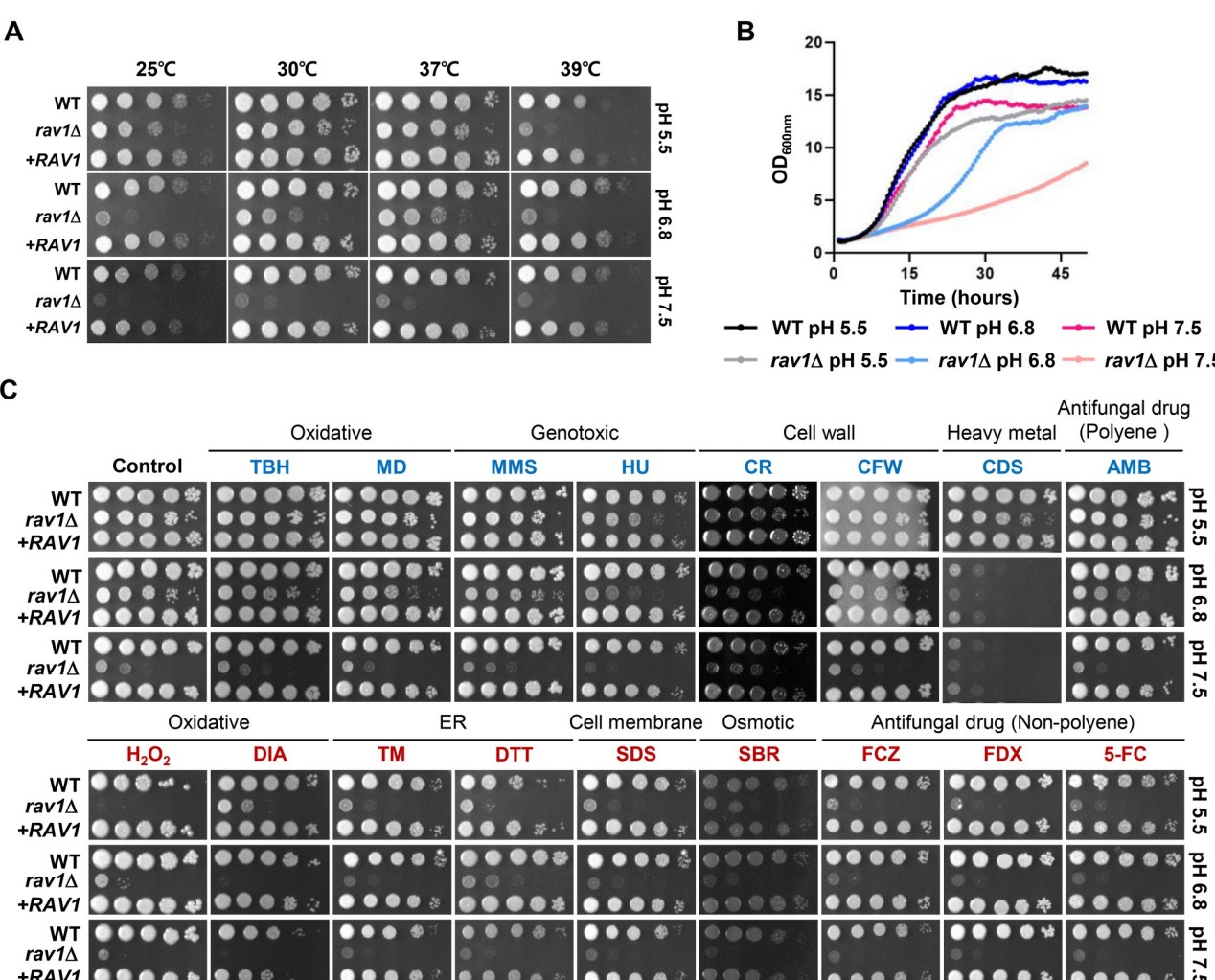

**Fig 6. Phenotypic analysis of pH-dependent or -independent function of the hRAVE complex.** (A) Qualitative spot assays under various temperatures (25˚C, 37˚C, and 39˚C) and pH levels (pH 5.5, pH 6.8, and pH 7.5). (B) Quantitative growth rates of the WT and *rav1Δ* strains on different pH levels. Growth was monitored at 30˚C by measuring OD$_{600}$ with a multi-channel bioreactor for 50 h. Each curve represents a single trial of two independent experiments. (C) Qualitative spot assays under indicated stress conditions at different pH levels. Plates were incubated at 30˚C for 3 days. Abbreviations: TBH (tert-butyl hydroperoxide), 0.7 mM; MD (menadione), 0.02 mM; MMS (methyl methanesulphonate), 0.03%; HU (hydroxyurea), 100 mM; CR (Congo red), 0.8%; CFW (calcofluor white), 5 mg/ml; CDS (cadmium sulfate), 25 μM; AMB (amphotericin B), 1.6 μg/ml; H$_2$O$_2$ (hydrogen peroxide), 3 mM; DIA (diamide), 2.5 mM; TM (tunicamycin), 0.3 μg/ml; DTT (dithiothreitol), 16 mM; SDS (sodium dodecyl sulfate), 0.03%; SBR (YPD + 2 M sorbitol); FCZ (fluconazole), 10 μg/ml; FDX (fludioxonil), 1 μg/ml; 5FC (5-flucytosine), 300 μg/ml. Blue and red fonts indicate pH-dependent and -independent phenotypes, respectively.

influence on growth, morphogenesis, sexual differentiation, melanin production, and various stress responses and adaptations. To establish a direct association with V-ATPase, we treated wild-type with a specific V-ATPase inhibitor, bafilomycin A1. The concentration of bafilomycin A1 utilized in this study was 500 nM, which has been previously reported to specifically inhibit V-ATPase at this dose [41,42]. By treating the wild-type strain with this inhibitor, we aimed to determine if it could lead to *rav1Δ*-like phenotypic changes in wild-type *C. neoformans*. First, bafilomycin A1 can cause defects in the growth of *C. neoformans* at the temperature range of 25˚C to 39˚C (Fig 7A). Second, bafilomycin A1 treatment caused elongated, pseudohyphae-like cellular morphology in wild-type *C. neoformans* in a dose-dependent manner (Fig 7B). Third, bafilomycin A1 treatment inhibited mating of wild-type *MAT*α and

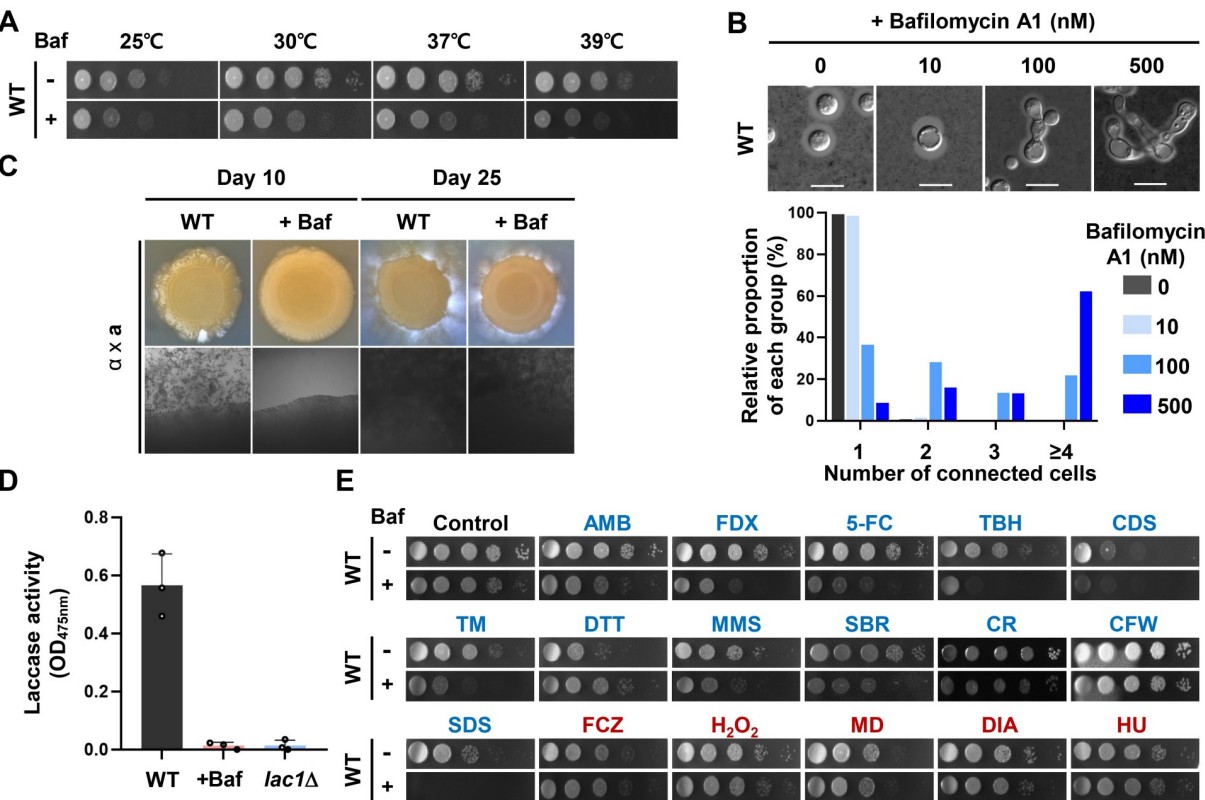

**Fig 7. Phenotypic analysis of V-ATPase-dependent or -independent function of the hRAVE complex.** (A-E) Wild-type phenotypic traits in response to bafilomycin A1 treatment (500 nM). (A) Temperature-dependent stress response of wild-type. The plates were incubated at 30°C for 2 days. (B) Morphological change of wild-type strain. The capsule was stained with India ink and observed by microscopy (Bar, 10 μm). More than 50 cells were observed and photographed. (C) Mating assay for bafilomycin A1 treatment in early and late phases. α H99 × **a** YL99 strains were cocultured on V8 medium with or without bafilomycin A1 for 10 to 25 days at room temperature in the dark. One representative from three biological replicates is shown here. (D) Laccase assay for bafilomycin A1 treatment. WT (H99) and *lac1*Δ (CHM3) mutants were incubated for 4 days in the YNB medium (see Fig 4B) supplemented with bafilomycin A1. (E) Phenotypic traits of WT treated with bafilomycin A1 under stress conditions described in Fig 6C. The plates were incubated at 30°C for 3 days. Blue and red fonts indicate V-ATPase-dependent and -independent phenotypes, respectively.

*MAT***a** strains at 10 days postmating on V8 medium, although its inhibition effects disappeared at 25 days postmating (Fig 7C). Fourth, bafilomycin A1 treatment completely abolished melanin production in *C. neoformans*, which is almost equivalent to the effect of *LAC1* deletion (Fig 7D). All these results indicate that the hRAVE complex plays pleiotropic roles in growth, morphogenesis, sexual development, and melanin production in *C. neoformans* via V-ATPase-dependent pH regulating functions.

Finally, we examined the effect of bafilomycin A1 on the resistance of *C. neoformans* against various stress-inducing agents. In agreement with the data presented in Fig 6C, the effect of bafilomycin A1 treatment varied with different kinds of stress-inducing agents. Under most stresses, bafilomycin A1-treated wild-type cells exhibited reduced growth rates compared to controls, indicating that the hRAVE complex is involved in stress response and adaptation in a predominantly V-ATPase-dependent manner. On the other hand, upon exposure to several stressors, including fluconazole, $H_2O_2$, menadione, diamide, and hydroxyurea, the growth rate of the wild-type strain treated with bafilomycin A1 exhibited no significant difference from the control group (Fig 7E). Notably, the increased susceptibility of *rav1*Δ to fluconazole and $H_2O_2$ also appeared to be pH-independent phenotypes (Fig 6C), suggesting that the hRAVE complex

may play a novel role in stress response and adaptation beyond its well-established function as a regulator of V-ATPase activity. To further elucidate the V-ATPase-dependency of the hRAVE complex, we conducted the same experiments on the *rav1Δ* mutant (S8A Fig). In most V-ATPase-dependent phenotypes, the growth rate of the *rav1Δ* mutant treated with bafilomycin A1 showed no significant difference from the untreated control, indicating that *rav1Δ* plays a crucial role as a regulator of V-ATPase assembly. However, under certain stresses, such as amphotericin B, tert-butyl hydroperoxide, calcofluor white, and hydroxyurea, we observed further growth inhibition in the *rav1Δ* mutant upon bafilomycin A1 treatment. This suggests the possible involvement of other regulators in V-ATPase assembly under these specific stresses in *C. neoformans*.

Furthermore, our findings corroborate that these V-ATPase-independent stress responses are modulated either through a Wdr1-dependent pathway, as observed with menadione or diamide treatment, or via Wdr1-independent pathway, evident with fluconazole, hydrogen peroxide, or hydroxyurea treatment (S8B Fig). Therefore, in the context of V-ATPase-independent stress responses, Rav1 and Wdr1 may collaborate or Rav1 can function independently. All these data indicate that the hRAVE complex plays V-ATPase/pH regulation-dependent and -independent roles in *C. neoformans*.

## Proteomics analysis for identifying hRAVE complex-interacting proteins

The finding that the hRAVE complex plays V-ATPase/pH-dependent and -independent roles in *C. neoformans* prompted us to search for other hRAVE-interacting partners. To this end, we performed *in vivo* pull-down experiment using the *rav1Δ::RAV1-mCherry* strain and RFP-trap agarose beads to precipitate Rav1-mCherry protein and its interacting partners. As expected, Wdr1 (CNAG_02402) was identified as one of the Rav1-interacting proteins (S2 Table), further supporting that Rav1 and Wdr1 likely constitute the hRAVE complex, as we predicted. Furthermore, one of the V-ATPase subunits (Vma1; CNAG_02326) was also identified (S2 Table), suggesting that the hRAVE complex interacts with V-ATPase. Interestingly, Rav1 appeared to interact with α (CNAG_05750) and β (CNAG_05918) subunits of F-type ATPase, suggesting that the hRAVE complex could control other ATPases.

V-ATPase has been reported to interact with several glycolytic enzymes, including aldolase, phosphofructokinase, and glyceraldehyde-3-phosphate dehydrogenase [43,44]. We also found that several glycolysis-related proteins were potential Rav1 interactors in *C. neoformans*: fructose-bisphosphate aldolase (CNAG_06770), glyceraldehyde-3-phosphate dehydrogenase (CNAG_06699), phosphoglycerate kinase (CNAG_03358), phosphopyruvate hydratase (CNAG_03072), pyruvate kinase (CNAG_01820), pyruvate decarboxylase (CNAG_04659). Furthermore, functional categories of Rav1-interacting proteins include amino acid biosynthesis: 5-methyltetrahydropteroyltriglutamate-homocysteine S-methyltransferase (Met6; CNAG_01890) and aspartate carbamoyltransferase (CNAG_07373). Stransky et al. demonstrated that amino acids regulate the activity and assembly of V-ATPase [45], indicating that the methionine and aspartate synthesis pathway may function with the RAVE complex. All these data implied that the hRAVE complex could interact with glycolytic and amino acid biosynthetic enzymes to regulate V-ATPase assembly.

Besides the Rav1-interacting proteins implicated in the V-ATPase-dependent roles of the hRAVE complex, the following Rav1-interactors appeared to be involved in the V-ATPase-independent functions of the hRAVE complex. Transaldolase (Tal1; CNAG_01984) was identified to be one of the Rav1-interacting proteins (S2 Table). Tal1 is involved in the pentose phosphate pathway in yeast [46] and controls oxidative stress response and mitochondrial function in humans [47]. Consistent with these data, the

deletion of *RAV1* exhibited severe V-ATPase-independent defects on oxidative stress (Fig 6C). These proteomics data support the result of novel functions of the V-ATPase-independent hRAVE complex and indicate that specific stress responses may be modulated through Rav1-Tal1 interactions.

## Roles of the hRAVE complex for the pathogenicity of *C. neoformans*

To investigate the role of the hRAVE complex in the pathogenicity of *C. neoformans*, we used wild-type, *rav1Δ*, and complemented strains in insect and murine models of systemic cryptococcosis. Our results showed that the virulence of the *rav1Δ* mutant was significantly attenuated compared to the wild-type and complemented strains in both insect (Fig 8A) and murine host models (Fig 8B). Further analysis of fungal burden in infected organs recovered from sacrificed mice revealed that all the *rav1Δ*-infected organs had reduced fungal burden compared to the wild-type and complemented strain-infected organs (Fig 8C). Additionally, the histopathological analysis showed that *rav1Δ* cells penetrated the lung tissue less efficiently than the wild-type and complemented cells, with the former observed in a clustered form inside the lung's bronchioles (Fig 8D). Our finding suggest that Rav1 is essential for *C. neoformans* to survive, proliferate and invade lung tissues effectively during infection, and may control

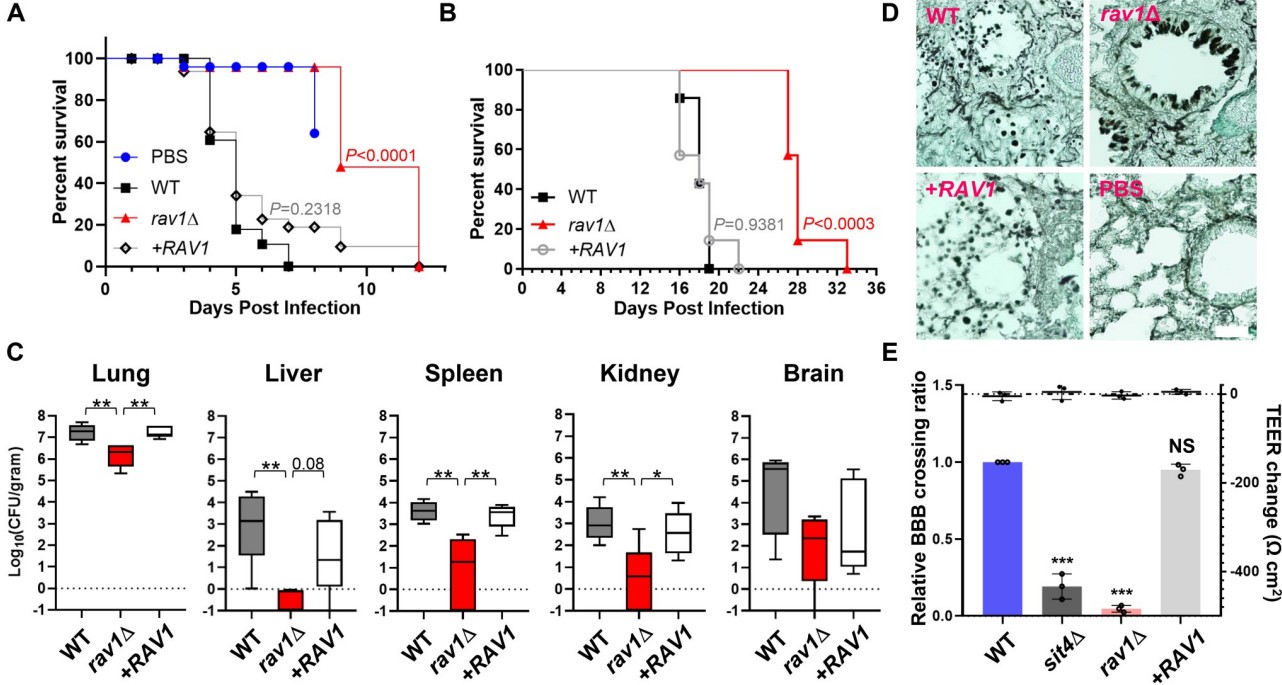

**Fig 8. The role of Rav1 in the pathogenicity of *C. neoformans*.** (A) *Galleria mellonella* insect killing assay (n>15) and (B) survival assay in BALB/c murine infection model (n = 7). Statistical differences between the WT (H99) and *rav1Δ* strain or between WT and *+RAV1* complemented strains were calculated using the log-rank (Mantel–Cox) test to measure. (C) Fungal burden assay (n = 5). Mice were sacrificed on day 13 post-infection. Statistical differences were calculated using the one-way ANOVA test. (D) Grocott's methenamine silver (GMS) staining of lung tissues 14 days after infection. The scale bar indicates 20 μm. (E) BBB crossing assay. The number of yeast cells that either transcytosed through or did not transcytose through the hCMEC/D3-coated Transwell was quantified using colony-forming units (CFU) for each category. The BBB crossing efficiency was calculated as outlined in the Materials and Methods section. The left Y-axis represents the relative BBB crossing ratio, which indicates the normalized BBB crossing efficiency for each tested strain relative to the wild-type (WT) strain. The right Y-axis displays the trans-endothelial electrical resistance (TEER). *C. neoformans sit4Δ* mutant were used as negative controls. Statistical significance was determined using one-way ANOVA with Tukey's multiple-comparison test: *, $P < 0.05$; **, $P < 0.01$; ***, $P < 0.0001$; NS, not significant. Error bars indicate SEM.

virulence by altering the ability to transverse the blood-brain barrier (BBB). Indeed, the *rav1Δ* mutant completely lost the ability to cross the BBB ([Fig 8E]). In addition, *wdr1Δ* and *wdr1Δ rav1Δ* mutants also exhibited severe defects in BBB crossing ([S9 Fig]), indicating that the hRAVE complex is critical for BBB transmigration. These results suggest that the hRAVE complex is required for the pathogenicity of *C. neoformans*.

## Discussion

In this study, we identified and characterized the cryptococcal RAVE (hRAVE) complex, a hybrid form of yeast RAVE and human Raboconnectin-3 complexes that are central V-ATPase assembly regulators for organelle acidification and vesicle trafficking in eukaryotes. Here we demonstrated that the hRAVE complex plays pleiotropic roles in the growth, morphogenesis, sexual reproduction, melanin production, diverse stress responses and adaptations, and pathogenicity of *C. neoformans* via both V-ATPase-dependent and -independent mechanisms, which are summarized in [Fig 9].

Rav1, the major subunit of the hRAVE complex, shares a similar structure and size with its orthologs in Ascomycota, including *S. cerevisiae*, *C. albicans*, and *S. pombe*. In contrast, the minor subunit of the hRAVE complex, Wdr1, consists of WD40 repeats, similar to higher eukaryotes but not Ascomycota. This pattern was also observed in other Basidiomycota species, including *Ustilago maydis* and *Malassezia globosa*. Therefore, we speculate that the RAVE complex in Basidiomycota exists as a yeast-human hybrid, such as the hRAVE complex. It remains to be determined whether the RAVE complex of each Basidiomycota species has a

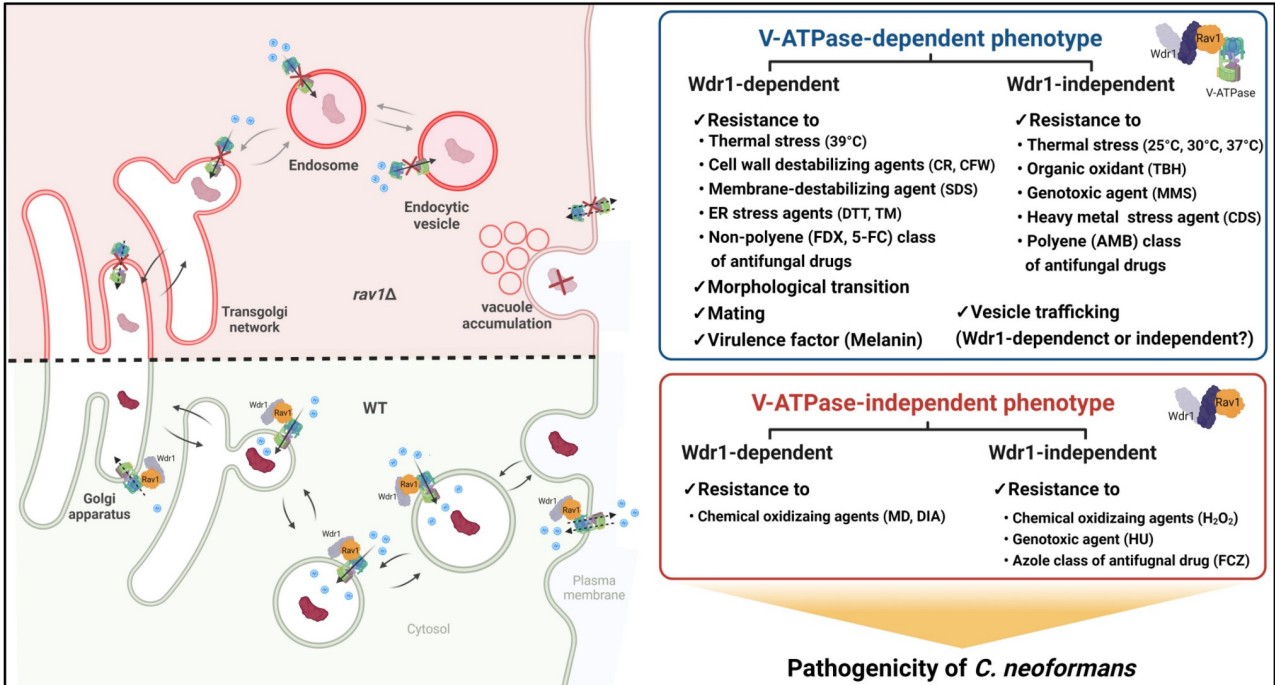

**Fig 9. Proposed pathobiological functions of the hRAVE complex in *C. neoformans*.** The hRAVE complex is critical for assembling the V-ATPase to maintain endosome acidity and transporting target proteins to their proper locations. It regulates growth, morphogenesis, mating, diverse stress responses and adaptations, virulence factor production, and pathogenicity of *C. neoformans* through both V-ATPase-dependent and -independent manners. Notably, Rav1, a subunit of hRAVE complex, operates in both Wdr1-dependent and -independent manners in *C. neoformans*.

similar function to the hRAVE complex of *C. neoformans* or if it has a distinct function that is unique in certain species.

The RAVE complex regulates the assembly of the V-ATPase complex, whose subunit composition exhibits structural diversity. The yeast $V_0$ subunit a isoform of V-ATPase exists in two types, Vph1 in vacuoles and Stv1 in Golgi and endosomes [48,49]. The expression level of Vph1 is significantly greater than that of Stv1 [50]. Vph1-containing complexes are more glucose-responsive than Stv1-containing V-ATPases [51]. Notably, it is known that the yeast RAVE complex directly binds with Vph1, but not with Stv1. Stv1-containing V-ATPases can nevertheless be assembled without the RAVE complex [24]. However, yeast $V_0$ subunit a isoforms, Vph1 and Stv1, have been identified as a single protein, CNAG_01106 (Vph1), in *C. neoformans*, suggesting that cryptococcal Vph1 may perform more integrated functions for acidification of vacuoles, Golgi, and endosomes. Furthermore, we utilized the FungiDB (https://www.fungidb.org/) to perform BLAST analyses on Basidiomycota, and the results consistently revealed the same pattern in 34 out of the 35 Basidiomycota species (S3 Table). Nevertheless, similar to *S. cerevisiae* [24], it is conceivable that *C. neoformans* might have RAVE-independent mechanisms for regulating V-ATPase assembly. In this context, the acidification process may be slower or less efficient in the absence of the hRAVE complex, but basal-level assembly may still occur without its involvement. Further investigations are warranted to explore the existence and significance of such RAVE-independent pathways in V-ATPase assembly in *C. neoformans*.

The unique structural differences between yeast RAVE and V-ATPase complexes and those of Basidiomycota RAVE and V-ATPase complexes led us to anticipate that the hRAVE complex would exhibit functional differences to yeast. As expected, the phenotypes of cryptococcal *rav1Δ* differ from those of *rav1Δ* in other model yeasts. At optimal growth conditions (30°C on YPD medium), where *S. cerevisiae rav1Δ* does not show any growth defects [20,24], cryptococcal *rav1Δ* exhibited severe growth defects. This disparity between non-pathogenic yeast and the pathogenic fungus *C. neoformans* suggests that the RAVE complex may play a more complex and critical function in the latter. Consistent with these hypotheses, our findings demonstrate that the hRAVE complex performs both V-ATPase-dependent and V-ATPase-independent roles in *C. neoformans*.

We have shown that the hRAVE complex has V-ATPase-dependent functions in growth and cell cycle regulation, morphogenesis, sexual development, melanin production, and stress responses. As previously stated, we confirmed that *rav1Δ* exhibited severe defects in melanin production caused by V-ATPase malfunction. However, the expression of the laccase gene increased in *rav1Δ*, indicating that the melanin production defect occurs after laccase synthesis. In agreement with the previous report showing that Rav1 is essential for vesicle trafficking in *S. cerevisiae* [35], we provided several lines of evidence showing that Rav1 has similar functions in *C. neoformans*. First, the cellular localization of Rav1 was highly enriched on the vacuolar membrane. Second, Rav1 is required for the mating-induced membrane localization of the pheromone transporter Ste6. In *S. cerevisiae*, Ste6 is transported to the cell surface but does not accumulate in the cell membrane; instead, it is returned to the vesicle for degradation via ubiquitination-dependent endocytosis [52]. Consequently, Ste6 is predominantly localized in vesicles [53]. In *C. neoformans*, Ste6 is localized in ER and vesicles under vegetative growth conditions but transported to the cell membrane under mating conditions, probably because the endocytic pathway of Ste6 is inhibited during mating. However, in cryptococcal *rav1Δ*, Ste6 was not transported to the membrane and accumulated in the vacuole during mating. If the hRAVE complex is only involved in Ste6 endocytosis, Ste6 should be accumulated on the cell membrane in *rav1Δ* even under non-mating conditions. Therefore, we suggest that the hRAVE complex is involved in bidirectional vacuole trafficking.

The role of the hRAVE complex in the vesicle trafficking of *C. neoformans* was further confirmed by proteomics analysis. Clathrin-mediated endocytosis is a crucial step in eukaryotic cell vesicle trafficking, which carries proteins from the cell surface to the interior [54]. Our proteomics analysis revealed that Rav1 could interact with Chc1, a heavy chain of clathrin that is the polymeric assembly unit of a vesicle coat in yeast [55–57]. It has been reported that the Chc1 protein is essential for endocytosis in yeast and *C. neoformans* [58]. In addition, *rav1Δ* and *chc1Δ* shared phenotypic traits, such as impaired morphogenesis, reduced growth at high temperatures, decreased production of virulence factors like capsule and melanin, and increased susceptibility to fluconazole and cell wall stress agents [58]. Multiple common phenotypes suggest that the hRAVE complex in *C. neoformans* could regulate vesicle trafficking of the early endosome by interacting with transport-related proteins and the clathrin-mediated endocytosis by interacting with Chc1. To elucidate the precise step, partners, and mechanism by which the hRAVE complex modulates vesicle trafficking, further research is required.

In this study, we noted that the hRAVE complex performs V-ATPase-independent functions. We found that extracellular acidic pH adjustment could not restore some stress sensitivity phenotypes in *rav1Δ* (pH-independent phenotypes). However, it is important to note that while V-ATPase activity is responsible for organelle acidification, merely altering the extracellular pH to acidic conditions may not suffice to fully restore V-ATPase function. For example, although extracellular acidic pH adjustment did not restore thermotolerance and resistance of *rav1Δ* to tunicamycin (an ER-stress agent), SDS (a membrane-destabilizing agent), and certain antifungal agents (fluconazole and flucytosine), treatment with the V-ATPase inhibitor bafilomycin A1 increased its thermosensitivity and stress and antifungal agent susceptibility. However, the role of the hRAVE complex in resistance to fluconazole and certain oxidative agents, such as $H_2O_2$ and diamide, appeared to be regulated in a V-ATPase-independent manner. $H_2O_2$ and fluconazole treatment commonly generate reactive oxygen species (ROS). $H_2O_2$ is a critical ROS against which *C. neoformans* must defend to maintain its virulence [59–61], and it has been reported that fluconazole increases the intracellular ROS in *C. neoformans* [62]. Therefore, the hRAVE complex may be involved in controlling the ROS level in a V-ATPase-independent manner. Supporting this, we found that Rav1 interacts with various enzymes and subunits associated with mitochondria, the primary source of ROS in fungi via oxidative phosphorylation [63]. These data suggest that the hRAVE complex may regulate mitochondrial oxidative phosphorylation and intracellular ROS levels by interacting with F-type ATPase subunits as well as diverse mitochondrial enzymes independently of V-ATPase. However, it remains to be further elucidated whether the hRAVE complex could regulate other ROS defense systems.

V-ATPase has long been regarded as a promising antifungal drug target [64]. This is due to the fact that even if one subunit of V-ATPase is deleted, it exhibits a severe Vma⁻ phenotype [3]. Developed examples of V-ATPase-targeting agents include bafilomycin A1 and concanamycin A [65]. However, since V-ATPase is highly conserved among eukaryotes, developing antifungal medications that can differentiate between human and fungal V-ATPase would be challenging [65,66]. Instead, due to its unique structural characteristics, the hRAVE complex, an assembly regulator of V-ATPase, could be more effectively exploited as an anticryptococcal target than V-ATPase itself. In a murine model of systemic cryptococcosis, deletion of *RAV1* significantly reduced the virulence and fungal burden of *C. neoformans*. This was due to an apparent delay in fungal invasion into the lungs and brain (Fig 8). Cytosolic pH regulation is important for fungal pathogenicity, and it is known to regulate filamentation, which is involved in the host tissue invasion of *C. albicans* [67]. Furthermore, vacuolar acidification is associated with the expression of adhesion and invasion membrane proteins [68,69]. Therefore, it is very promising to develop highly specific protein inhibitors that control the tissue

invasion and pathogenicity of fungi. Moreover, the hRAVE complex consists of WD40 repeats-containing proteins Rav1 and Wdr1. The WD40 domain regulates protein-protein interactions using its structural properties [70]. When WD40 proteins form a β-propeller structure, a central cavity can be used to regulate protein-protein interactions. This space has a high degree of specificity, and if an inhibitor can be designed to suit it, it could be a highly effective agent [71]. Therefore, the hRAVE complex, which is composed of WD40 proteins with potential drug targets, can be viewed as a prospective drug target.

## Materials and methods

### Ethics statement

Animal care and research were approved after deliberation by the Institutional Animal Care and Use Committee of the Experimental Animal Center at Jeonbuk National University. (Approval number JBNU 2022–092) All experiments followed the experimental ethics guidelines.

### Strains, media, and growth conditions

*C. neoformans* strains used in this study are listed in S4 Table in the supplemental material. Strains were maintained and cultured in 1% yeast extract-2% peptone-2% dextrose (YPD) broth medium or on plates containing 2% agar in YPD broth. Niger seed, $_L$-3,4-dihydroxyphenylalanine ($_L$-DOPA), or epinephrine agar medium for melanin production assay and liquid epinephrine medium for laccase assay; agar-based Dulbecco's modified Eagle (DME), Roswell Park Memorial Institute (RPMI) 1640, and Littman's media for capsule production assay; and V8 medium containing 5% V8 juice for mating were made as previously described [72].

### Construction of *rav1Δ* and *wdr1Δ* mutants and complemented strains

To generate *rav1Δ* and *wdr1Δ* mutants, we employed homologous recombination in the *C. neoformans* serotype A *MAT*α wild type strain (H99) and *MAT***a** strain (YL99**a**) backgrounds, respectively. We used gene disruption cassettes containing the nourseothricin resistance marker (*NAT*) and neomycin resistance marker (*NEO*). Mutants were constructed according to previously described methods [73]. Targeted gene deletion was confirmed by Southern blot analysis (see S2 Fig in the supplemental material). To validate the observed phenotypes of the *rav1Δ* mutant, we constructed non-tagging complemented strain via the Gibson assembly method. The full-length gene fragment was amplified via Phusion PCR using the H99 genomic DNA as the template. The amplified fragments of the *MAT*α *RAV1* gene were cloned with pNEO plasmid. To observe the localization of Rav1, we also constructed fluorescence (mCherry)-tagged complemented strain by cloning the amplified *RAV1* insert into pNEO_mCherry plasmid. After confirming the integration of the target gene into each plasmid via enzyme digestion and sequencing analysis, targeted reintegration into the native locus was performed through biolistic transformation. Plasmids containing the *RAV1* gene were linearized by restriction enzyme digestion (NdeI) and introduced into the *rav1Δ* mutant strain through biolistic transformation. The correct targeted integration of the complemented strains was confirmed via diagnostic PCR.

### Construction of heterozygous *SKP1/skp1Δ* mutants and random spore analysis

For the generation of heterozygous *SKP1/skp1Δ* mutants, we utilized homologous recombination techniques in the genetically engineered diploid *C. neoformans* strain AI187 (*ade2/ADE2*

*ura5/URA5 MAT***a**/*MAT*α). Gene disruption cassettes containing the nourseothricin resistance marker (*NAT*) were used. The knockout construction and validation protocol follow those established for haploid knockout strains. Targeted gene deletion was verified through Southern blot analysis (see S1E Fig in the supplemental material). For sporulation induction, the constructed *SKP1/skp1*Δ mutants were cultured overnight in liquid YPD medium at 30˚C. We spotted 3 μl of $10^7$ cells/ml onto V8 mating medium (pH 5) and incubated the plates in the dark at room temperature (25˚C) for over 30 days. For spore isolation, we followed the protocol previously reported by Frerichs et al. [74] with a slight modification. Following incubation on the mating medium, spores were harvested by scraping and resuspended in 75% Percoll buffer (Sigma-Aldrich, P1644). Spore suspensions were vigorously vortexed for 10 seconds until fully resuspended. Subsequently, samples were centrifuged at 3,000 rpm (1,977 x *g*) at 4˚C for 30 min. The bottom of the tube was sterilized with 70% ethanol, and a needle was employed to puncture it. Five drops were collected, followed by the addition of 1.2 ml of PBS. Tubes were centrifuged again for 5 min at 3,000 rpm, and the supernatant was discarded, retaining 100 μl of buffer. An additional 1.4 ml of PBS was added for washing, and this step was repeated twice. For spore plating and verification, spores were then diluted at a 1/100 ratio, and 50 μl of this diluted spore suspension was plated onto YPD medium supplemented with 5-fluoroorotic acid (5-FOA; 100 μg/ml), adenine (20 μg/ml), and uracil (40 μg/ml). Plates were incubated at 30˚C for 3 days, after which emerging colonies were streaked onto YPD plates containing chloramphenicol. After 2 days, isolated spores were spotted on various media, including YPD, YPD with nourseothricin (100 μg/ml), YNB, YNB with uracil, YNB with adenine, and YNB with both uracil and adenine. To confirm the haploid status and mating types, diagnostic PCR targeting the mating locus was performed on isolated spores. For those grown in nourseothricin-supplemented YPD, diagnostic PCR was conducted to confirm the successful internal gene deletion.

### Measurement of the number of connected cells

Each strain was incubated overnight in liquid YPD medium at 30˚C. The cells were washed twice with PBS, resuspended in 0.5 ml of PBS, and spotted (3 μl) onto YPD, Littman, or RPMI agar media. The plates were incubated at 37˚C for 2 days. After incubation, a single colony was selected from each plate and suspended in PBS. The cells were then washed twice in PBS and sonicated using a VCX 130 (6mm tip; SONICS, Japan) with a CV18 converter (Vibra-Cell). Sonication was performed for 10 seconds at an amplitude of 20%. Capsule visualization was performed by staining with India ink, and morphology was observed by microscopy (Bar, 10 μm). Over 50 cells were observed and photographed for each wild-type and mutant strains.

### Growth and stress sensitivity assays

Each strain was incubated overnight in liquid YPD medium at 30˚C with shaking at 220 RPM. The cultures were then serially diluted (1 to $10^4$ dilutions) in distilled water (dH$_2$O), and spotted (3 μl) onto a solid YPD medium. To assess the growth of *C. neoformans* strains at different temperatures, serially diluted cells were spotted on YPD agar medium and incubated at 25˚C, 37˚C, and 39 ˚C for 1–5 days, with daily photographs taken. To quantitatively examine the growth rate of the *rav1*Δ mutant, the WT strain (H99) and *rav1*Δ mutant were incubated at 30˚C overnight and subcultured in a fresh liquid YPD medium [optical density at 600 nm ($OD_{600}$) = 0.2]. The cells were then incubated in a multi-channel bioreactor (Biosan Laboratories, Inc., Warren, MI, USA) at 30˚C, 37˚C, and 39˚C, and $OD_{600}$ was automatically measured for 40–50 h. To investigate the response of *rav1*Δ mutant to various stresses, we spotted onto the YPD medium containing the indicated concentration of the following chemical agents:

antifungal drug (amphotericin B, fludioxonil, fluconazole, and flucytosine); oxidative stress (hydrogen peroxide [$H_2O_2$], tert-butyl hydroperoxide [an organic peroxide], menadione [a superoxide anion generator], diamide [a thiol-specific oxidant]); osmotic stress (sorbitol) under glucose-rich (YPD) condition; genotoxic stress (methyl methanesulfonate and hydroxy-urea); cell wall destabilizing stress (calcofluor white and Congo red); membrane destabilizing stress (sodium dodecyl sulfate [SDS]); ER stress (tunicamycin and dithiothreitol [DTT]); heavy metal stress (cadmium sulfate [$CdSO_4$]). Cells were incubated at 30°C for 1–5 days with daily photographs taken. To analyze stress responses according to the pH of the $rav1\Delta$ mutant, the same stress agents were added to YPD media adjusted to pH 5.5, 6.8, and 7.5 using 50 mM MES (for pH 5.5) or 150 mM HEPES (for pH 6.8 or 7.5).

### *In vitro* virulence factor production assays

For melanin production assay, each strain was incubated for 16 h at 30°C in YPD medium, washed three times with PBS, and spotted onto Niger seed, L-DOPA, or epinephrine agar medium containing 0.1% glucose. The cells were incubated at 37°C for 3 days with daily monitoring and photography of melanin production. Laccase activity was measured quantitatively, as previously described [75]. Briefly, each strain was incubated in 50 ml of YPD medium at 30°C for 16 h. The incubated cells were washed three times with PBS, adjusted to an $OD_{600}$ of 0.2, and inoculated into YNB plus 0.05% glucose supplemented with 10 mM epinephrine. The cultures were incubated at 30°C for 4 days. After incubation, cells were centrifuged, and the $OD_{475}$ of the supernatant was measured with a spectrophotometer. For capsule production assay, each strain was incubated in liquid YPD medium at 30°C for 16 h, spotted onto agar-based DME, Littman, or RPMI medium, and further incubated for 48 h at 37°C. Capsule production was observed by staining with India ink and examining it under a microscope. Microscopy images were captured using a Nikon Eclipse Ni microscope. Unfortunately, quantitative measurement of capsule size was not possible due to the abnormal morphology of the $rav1\Delta$ mutant.

### Mating and cell fusion assays

Mating and cell fusion efficiency was measured as previously described [76]. For unilateral mating, equal concentrations ($10^7$ cells/ml) of $rav1\Delta$ mutant cells of one mating type were mixed with wild-type cells of the opposite mating type. For bilateral mating, equal concentrations ($10^7$ cells/ml) of $rav1\Delta$ mutant cells of both mating types were mixed. These mixtures were spotted onto a V8 mating medium (pH 5) and incubated in the dark at room temperature (25°C) for 7 to 25 days. Filamentous growth was monitored using a differential interference contrast (DIC) microscope (BX51; Olympus, Japan) and photographed using an Olympus BX51 microscope equipped with a SPOT Insight digital camera (Diagnostic Instruments, Inc.). For the cell fusion efficiency assay, each *MAT*α and *MAT***a** strain bearing *NAT* or *NEO* markers, respectively, was mixed in an equal concentration ($10^7$ cells/ml) and spotted (5 μl) onto the V8 medium. The cells were then incubated in the dark at room temperature for 24 or 48 h. After scraping and resuspending the cells, they were 100-fold diluted in distilled water, and 200 μl of each sample was spread onto a YPD medium containing both nourseothricin (100 μg/ml) and neomycin (50 μg/ml). After incubation in the dark at room temperature for 4 to 5 days, the number of colonies on each plate was counted using an automated bacterial colony counter (aCOLyte 3; Synbiosis Ltd., UK).

### Total RNA preparation and quantitative RT-PCR

WT and *RAV1* deleted strains were grown in 50 ml YPD broth at 30°C for 16 h. The cultures were then subcultured into 100 ml of fresh YPD broth and incubated until the $OD_{600}$ reached

0.8. At this point, the cells were harvested by centrifugation, frozen in liquid nitrogen for at least 30 min, and lyophilized. For stress response experiments, 50 ml of the culture was sampled as the zero time-point (basal condition), and the remaining 50 ml was treated with the indicated stress agents or conditions. Total RNAs were extracted as previously described [77], except using easy-Blue Total RNA extraction kit (iNtRON) instead of TRIzol extraction solution. The purified total RNA was then used for cDNA synthesis by reverse transcriptase (Thermo Scientific), and the resulting cDNA was used for quantitative PCR with specific primer pairs for each gene, using the CFX96 real-time system (Bio-Rad). Relative gene expression levels were monitored using the gene-specific primers listed in S5 Table.

## Transmission electron microscopy

To prepare samples for electron microscopy analysis, WT and $rav1\Delta$ strains were cultured overnight at 30˚C in the liquid YPD medium, and then sub-cultured to $OD_{600}$ of 0.8. One milliliter of cells was fixed for 12 h in 2% glutaraldehyde-2% paraformaldehyde in 0.1 M phosphate buffer (pH 7.4), washed in 0.1 M phosphate buffer, and post-fixed with 1% $OsO_4$ in 0.1 M phosphate buffer for 2 h. The cells were dehydrated using an ascending ethanol series (50, 60, 70, 80, 90, 95, 100%) for 10 min each and infiltrated with propylene oxide for 10 min. Specimens were then embedded with a Poly/Bed 812 kit (Polysciences) and polymerized in an electron microscope oven (TD-700, DOSAKA, Japan) at 65˚C for 12 h. The block is equipped with a diamond knife in the Ultramichrome and is cut into 200 nm semi-thin sections, which were stained with toluidine blue for optical microscope observation. The region of interest was further cut into 80 nm thin sections using the ultramicrotome, placed on copper grids, double stained with 3% uranyl acetate for 30 min and 3% lead citrate for 7 min staining, and imaged with a transmission electron microscopy (JEM-1011, JEOL, Tokyo, Japan) at the acceleration voltage of 80 kV, equipped with a Megaview III CCD camera (Soft imaging system-Germany).

## Insect-based *in vivo* virulence assay

For *in vivo* insect virulence assays, we used at least 15 *Galleria mellonella* caterpillars (Vanderhorst Wholesale, Inc., Saint Marys, OH, USA) in the final larval instar with a body weight of 200–300 mg, delivered within 7 days from the shipment date. Each wild-type H99, $rav1\Delta$, and its complemented strain was incubated at 30 ˚C overnight, pelleted, washed three times with PBS, and resuspended in PBS at a $10^6$ cells/ml concentration. Four thousand *C. neoformans* cells per larvae were injected into the second-to-last prolegs of each larva using a 100-μl syringe equipped with a 10-μl needle and repeating dispenser (PB600-1, Hamilton Company, Reno, NV, USA). Negative control *G. mellonella* received PBS only. Infected larvae were placed in Petri dishes in a humidified container, incubated at 37 ˚C, and monitored daily. Larvae were considered dead when they turned black and showed no movement upon touching. Larvae that were pupated during the experiment were censored for statistical analysis. Prism 8 (GraphPad, San Diego, CA, USA) was used to illustrate survival curves, which were analyzed with a log-rank (Mantel–Cox) test.

## Murine-based *in vivo* virulence assay

Animal experiments were conducted at the Core Facility Center for Zoonosis Research (Jeonbuk National University, South Korea). SPF/VAF-confirmed inbred 6-week-old female BALB/cAnNCrlOri mice were purchased from ORIENT BIO INC.(South Korea) and acclimatized to the breeding environment for one week before the experiment. For infection, strains were inoculated in fresh liquid YPD medium and cultured overnight a 30˚C with shaking. The number of yeast cells was adjusted, and the mice were anesthetized with isoflurane before the

strain was inhaled nasally. Mice were monitored daily for their condition, and survival rates were expressed as percent survival. For survival and histochemistry assay, $5 \times 10^5$ cells/mouse was inhaled nasally, and monitored. Statistical analysis was performed using the log-rank (Mantel–Cox) test. For the fungal burden assay, $3 \times 10^6$ cells/mouse was inhaled nasally, and lungs, liver, kidneys, spleen, and brain were collected on day 13 post-infection. Tissue weights were measured, and the colony-forming unit was determined by spreading the homogenized tissue on a YPD plate. The statistical significance of difference was determined using one-way ANOVA with Tukey's multiple-comparison test. GraphPad Prism 9.5.1 was used for statistical analysis.

## Histochemistry assay

For tissue staining, lungs were obtained from mice on day 14 after infection and fixed in 3.7% formalin solutions. After complete fixation, the tissue was dehydrated and clarified using xylene before being embedded in paraffin blocks. Tissue sections with a thickness of 5 μm were cut and stained with Grocott's Methenamine Silver (GMS) Stain Kit (ab287884, abcam, UK), following the manufacturer's protocol. The slide images were taken with a Primostar 3 (ZEISS, Germany) at a final 400× magnification.

## *In vitro* BBB crossing assay

The BBB crossing assay was performed as previously described [78] with modifications using EndoGRO-MV complete media (Merck Millipore). Briefly, $5 \times 10^4$ hCMEC/D3 cells in Endo-GRO-MV medium supplemented with 1 ng/ml of FGF-2 were seeded on collagen (Corning) coated 8 μm-porous membranes (BD Falcon). The day after seeding, the medium was replaced with a fresh EndoGRO-MV medium supplemented with 2.5% human serum and grown for 4 days. A day before yeast inoculation, the medium was replaced with a new EndoGRO-MV medium supplemented with 1.25% human serum, and the cells were maintained at 37 ˚C and 5% $CO_2$ incubator. The integrity of tight junctions between hCMEC/D3 cells was confirmed by measuring the trans-endothelial electrical resistance (TEER), which should be around 200 Ω per $cm^2$. TEER was measured by Epithelial Volt per Ohm Metre ($EVOM^2$ device, World Precision Instruments). For BBB crossing assay, $5 \times 10^5$ cells of *C. neoformans* WT (H99), *rav1Δ*, *wdr1Δ*, *wdr1Δ rav1Δ* mutants, and the *rav1Δ::RAV1-mCherry* strain were added to 100 μl of PBS and inoculated onto the top of the porous membranes. After 24 h incubation at 37 ˚C in a 5% $CO_2$ incubator, the number of yeast cells passing through the porous membrane was measured by counting CFU, and TEER was measured using the $EVOM^2$ device before and after inoculating yeast cells.

The BBB crossing ratio was determined using the following equation: the number of transcytosed CFU was divided by the total CFU (both transcytosed and non-transcytosed) for each tested strain. Subsequently, the ratio obtained for each tested strain was normalized to the corresponding value for the wild-type (WT) value.

$$BBB\ crossing\ ratio = CFU_{transcytosed}/CFU_{total\ (transcytosed\ +\ non-transcytosed)}$$

## Preparation for LC-MS/MS

Protein extraction methods were used in the previous study [79]. Rav1-mCherry strain was grown in 50 ml of YPD broth at 30˚C for 16 h, subcultured into one litter of fresh YPD broth, and incubated until the culture reached an $OD_{600}$ of 0.8. At this point, the cells were harvested by centrifugation, frozen in liquid nitrogen for more than 30 min, and then lyophilized. To

extract total proteins, the lyophilized cells were treated with lysis buffer (without SDS) containing 50 mM Tris-Cl (pH 7.5), 1% sodium deoxycholate, 5 mM sodium pyrophosphate, 0.2 mM sodium orthovanadate, 50 mM sodium fluoride (NaF), 1% Triton X-100, 0.5 mM phenylmethylsulfonyl fluoride, and 2.5 × protease inhibitor cocktail solution (Merck Millipore). Total proteins were extracted as described previously [80]. The RFP-Trap Agarose (ChromoTek, USA) was added to the extracted whole cell lysates and incubated overnight while rotating at 4˚C. Proteins bound to the agarose beads were washed three times and eluted with SDS sample buffer (50 mM Tris-Cl, 2% SDS, 10% glycerol, and 0.01% β-mercaptoethanol). Protein samples were loaded into 8% SDS-PAGE gel and run until the total band migration was about 1 cm. The gels were washed with distilled water and stained with the Coomassie Brilliant Blue G250 dye (CBB; Biosesang) overnight. The gel was washed three times with distilled water, and the CBB destaining buffer (10% methanol, 10% acetic acid, 80% distilled water) was added to remove any remaining coomassie blue buffers. The gel spots of interest for LC-MS/MS analysis were excised from the preparative gel and transferred into each 1.5 ml tube. The band was washed with 100 μL of distilled water; then, 100 μL of 50 mM $NH_4HCO_3$ (pH 7.8) and acetonitrile (6:4) were added to the band and shaken for 10 min. This process was repeated at least three times until the Coomassie brilliant blue G250 dye disappeared. The supernatant was decanted, and the band was dried in a speed vacuum concentrator (LaBoGeneAps, Lynge, Denmark) for 10 min. The solution was decanted and then digested with sequence-grade modified trypsin (Promega Co., Madison, WI, USA) (enzyme to substrate ratio = 1:50) at 37 ˚C with shaking for 16 h.

## LC-MS/MS for peptides analysis

Nano LC-MS/MS analysis was performed with a Easy n-LC (Thermo Fisher San Jose, CA, USA). The capillary column used for LC-MS/MS analysis (150 mm × 0.075 mm) was obtained from Proxeon (Odense, Denmark) and the slurry packed in-house with a 5 μm, 100 A pore size Magic C18 stationary phase resin (Michrom BioResources, Auburn, CA). The mobile phase A for LC separation was 0.1% formic acid in deionized water and the mobile phase B was 0.1% formic acid in acetonitrile. The chromatography gradient was designed for a linear increase from 3% B to 25% B in 70min, 25% B to 60% B in 13:59 min, 60% B to 95% B in 1:59 min, 95% B in 8:59 min, and 3% B in 6 min. The flow rate was maintained at 700 nl/min. LTQ-Orbitrap mass spectrometry (Thermo Fisher, San Jose, CA, USA) was used for the peptide identification. Mass spectra were acquired using data-dependent acquisition with a full mass scan (350–1200 m/z) followed by 10 MS/MS scans. For MS1 full scans, the orbitrap resolution was 15,000 and the AGC was $2\times10^5$. For MS/MS in the LTQ, the AGC was $1\times10^4$. The mascot server 2.6 (Matrixscience, USA) was used to identify peptide sequences present in a protein sequence database. Database search criteria were, database name; *C. neoformans*_H99_20210806 (15833 sequences; 8284938 residues), fixed modification; carbamidomethylated at cysteine residues; variable modification; oxidized at methionine residues, acetylated at N-term residue, deaminated at asparagine or glutamine residues, maximum allowed missed cleavage; 2, MS tolerance; 10 ppm, MS/MS tolerance; 0.8 Da. The peptides were filtered with a significance threshold of $P<0.05$. The mass spectrometry proteomics data have been deposited to the ProteomeXchange Consortium via the PRIDE [81] partner repository with the dataset identifier PXD042074 and 10.6019/PXD042074.

## Supporting information

**S1 Table. Human WDR7 orthologs in fungal species.**
(XLSX)

**S2 Table. Rav1-mCherry interacting proteins detected by LC-MS/MS analysis.**
(XLSX)

**S3 Table. Yeast *VPH1* and *STV1* orthologs in Basidiomycota.**
(XLSX)

**S4 Table. List of strains used in this study.**
(DOCX)

**S5 Table. List of primers used in this study.**
(DOCX)

**S1 Fig. *SKP1* gene is essential in *Cryptococcus neoformans*.** (A) The protein domain structure and phylogenetic tree of Skp1 in *C. neoformans*, *U. maydis*, *S. cerevisiae*, *S. pombe*, *C. albicans*, and *H. sapiens* are shown. (B) The various functions of yeast Skp1 were adapted from Seol et al. [20] and Kim et al. [82]. (C) A schematic strategy for the construction of promoter replacement mutant strains is presented. The P$_{CTR4}$:*SKP1* strains were generated in the haploid *C. neoformans* H99 strain background, and the correct genotypes of the mutants were confirmed by Southern blot analysis. (D) WT, P$_{CTR4}$:*TOR1* (YSB3176), and P$_{CTR4}$:*SKP1* (YSB10528-YSB10533) strains were spotted on YPD or YNB medium containing bathocuproine disulfonate (BCS; 200 and 300 μM) or 25 μM CuSO$_4$. The plates were incubated at 30° and photographed daily for 2 days. (E) A schematic strategy for the construction of heterozygous knockout mutant strains is presented. The *SKP1*/*skp1*Δ strains were generated in the diploid *C. neoformans* AI187 strain background, and the correct genotypes of the mutants were confirmed by Southern blot analysis. (F) The spores obtained through the sporulation assay strains were spotted on YPD and YPD+NAT (100 μg/ml), or YNB, YNB+URA (40 μg/ml), YNB+ADE (20 μg/ml), and YNB+URA+ADE medium. The plates were incubated at 30°C and photographed daily for 3 days. The deletion of the *SKP1* gene was verified using diagnostic PCR. A plus symbol indicates that the internal gene is still intact, suggesting the presence of aneuploidy and thereby ruling out the possibility of a knockout spore.
(TIF)

**S2 Fig. Construction of the *RAV1* complemented strain and *wdr1*Δ, *rav1*Δ, and *wdr1*Δ *rav1*Δ double deletion mutants.** (A) Targeted integration of non-tagged *RAV1* gene (*rav1*Δ:: *RAV1*) was confirmed by diagnostic PCR. (B) Disruption of the *WDR1*, *RAV1*, and both genes in the *MAT*α H99 strain and *WDR1* and *RAV1* in the *MAT***a** YL99 strain was confirmed by Southern blot using genomic DNAs digested with the indicated restriction enzymes. The representative strains used in this study are shown in bold.
(TIF)

**S3 Fig. The hRAVE complex functions independently of the RAM pathway in morphogenesis.** (A) Quantification of unseparated cells grown on YPD, DME, and Littman medium agar plates or liquid broth. Each *C. neoformans* strain was cultured at 37°C for 2 days and observed microscopically after brief sonication (>100 cell numbers). Shown are representative images of cells. (B) Morphological comparison between *rav1*Δ and *ckb1*Δ or *kic1*Δ in RAM pathway of *C. neoformans*. Each strain was cultured on YPD, YNB, Littman, DME, RPMI, and FBS agar media at 37°C for 2 days. ¼ YNB is a limiting nitrogen medium that contains 0.17% Yeast Nitrogen Base without amino acids or ammonium sulfate, 50 μM ammonium sulfate, 2% glucose, and 2% Bacto agar. Capsule was visualized by India ink staining. Scale bar indicates 10 μm.
(TIF)

**S4 Fig. Bilateral mating of *wdr1Δ* exhibited markedly reduced filamentation.** (A) Southern blot analysis confirmed the disruption of the *WDR1* and *RAV1* genes in the *MAT***a** YL99 strain. The representative strains used in this study are shown in bold and listed in S4 Table. (B) The indicated *MAT*α and *MAT***a** strains were cocultured in V8 medium plates (pH 5.0) for 25 days at room temperature in the dark and photographed on the indicated days. The strains used for the mating assay are as follows: α (H99) × **a** (YL99), α *wdr1Δ* × **a** (YL99**a**), α (H99) × **a** *wdr1Δ*, and α *wdr1Δ* × **a** *wdr1Δ*.
(TIF)

**S5 Fig. Quantitative RT-PCR analysis of capsule production-related genes and melanin production assay.** (A) The genes analyzed include *CAP10*, *CAP59*, *CAP60*, *CAP64*, *GAT201*, *YAP1*, and *ADA2*. Overnight cultures in YPD medium were subcultured to $OD_{600} = 0.8$ in fresh YPD medium. Each strain grown in YPD medium (time zero sample) was resuspended in Littman liquid medium, further incubated for 2 h, and extracted for total RNA. Each gene expression was normalized to *ACT1* expression. The statistical significance of difference was determined using one-way ANOVA with Bonferroni's multiple-comparison test: *, $P < 0.05$; **, $P < 0.01$; ***, $P < 0.0001$; NS, non-significant. Error bars indicate the standard error of the mean (SEM). (B) Melanin production assay. The wild-type (WT; H99), *cac1Δ* (YSB42), *rav1Δ* (YSB7589), *wdr1Δ* (YSB10032), and *wdr1Δ rav1Δ* (YSB10604) mutants were spotted onto L-DOPA agar medium containing 0.1% glucose, and incubated at 37˚C for 3 days, and then photographed.
(TIF)

**S6 Fig. Construction of *RAV1-mCherry* tagging strain.** (A) Targeted integration of *RAV1-mCherry* tagging strains was confirmed by diagnostic PCR. (B) Phenotypic traits of the *RAV1-mCherry* strain were evaluated. The wild-type (WT), *rav1Δ*, and *rav1Δ::RAV1-mCherry* strains were cultured overnight in YPD broth at 30˚C, serially diluted 10-fold, and spotted onto indicated stress conditions. The plates were incubated at 30˚C for 3 days.
(TIF)

**S7 Fig. Construction of *STE6-GFP rav1Δ* mutant strains.** Southern blot analysis confirmed the disruption of the *RAV1* in the *STE6-GFP* strain. Genomic DNA was digested with the indicated restriction enzymes. The representative strains used in this study are shown in bold.
(TIF)

**S8 Fig. Phenotypic analysis of the hRAVE complex.** (A) Phenotypic traits of *rav1Δ* treated with bafilomycin A1 under following stress conditions; AMB (amphotericin B), 0.5 μg/ml; FDX (fludioxonil), 0.005 μg/ml; 5-FC (5-flucytosine), 1 μg/ml; TBH (*tert-butyl* hydroperoxide), 0.7 mM; CDS (cadmium sulfate), 5 μM; TM (tunicamycin), 0.3 μg/ml; DTT (dithiothreitol), 0.5 mM; MMS (methyl methanesulfonate), 0.005%; SBR (YPD + 1 M sorbitol); CR (Congo red), 0.1%; CFW (calcofluor white), 3 mg/ml; SDS (sodium dodecyl sulfate), 0.0003%; FCZ (fluconazole), 1 μg/ml; $H_2O_2$ (hydrogen peroxide), 1 mM; MD (menadione), 0.01 mM; DIA (diamide), 0.1 mM; HU (hydroxyurea), 50 mM. (B) Phenotypic traits of *wdr1Δ* under following stress conditions; Temperature, 25, 30, 37, and 39˚C; 2.2 μg/ml AMB; 0.9 mM TBH; 25 μM CDS; 0.04% MMS; 25 μg/ml FCZ; 4 mM $H_2O_2$; 120 mM HU; SBR (YPD + 2 M sorbitol); 3 μg/ml FDX; 500 μg/ml 5-FC; 1.2% CR; 7 mg/ml CFW; 0.03% SDS; 18 mM DTT; 0.4 μg/ml TM; 2.5 mM DIA; 0.04 mM MD. Blue- and red-colored fonts indicate V-ATPase-dependent and -independent phenotypes, respectively.
(TIF)

**S9 Fig. BBB crossing assays of *rav1Δ*, *wdr1Δ*, and *wdr1Δ rav1Δ* using hCMEC/D3-coated Transwell systems.** The *in vitro* BBB Transwell system (hCMEC/D3-coated Transwell) was used to assess the transmigration of $10^5$ yeast cells for *rav1Δ*, *wdr1Δ*, and *wdr1Δ rav1Δ* mutants, incubated at 37 ˚C in a $CO_2$ incubator for 24 h. The BBB crossing efficiency was calculated as outlined in the Materials and Methods section. The left Y-axis represents the relative BBB crossing ratio, which indicates the normalized BBB crossing efficiency for each tested strain relative to the wild-type (WT) strain. The right Y-axis displays the trans-endothelial electrical resistance (TEER). The *sit4Δ* mutant was used as a negative control. The statistical significance of the differences was determined using one-way ANOVA with Tukey's multiple-comparison test: **, $P < 0.01$; ***, $P < 0.0001$. Error bars indicate SEM.
(TIF)

## Acknowledgments

We would like to thank Jeeseok Oh from Yonsei University, for his assistance in the gene knockout experiments.

## Author Contributions

**Conceptualization:** Yong-Sun Bahn.

**Data curation:** Jin-Tae Choi, Yeseul Choi, Yujin Lee, Seung-Heon Lee, Seun Kang, Yong-Sun Bahn.

**Formal analysis:** Jin-Tae Choi, Yeseul Choi, Yujin Lee, Seung-Heon Lee, Seun Kang, Kyung-Tae Lee, Yong-Sun Bahn.

**Funding acquisition:** Kyung-Tae Lee, Yong-Sun Bahn.

**Investigation:** Jin-Tae Choi, Yeseul Choi, Yujin Lee, Seung-Heon Lee, Seun Kang, Kyung-Tae Lee.

**Methodology:** Jin-Tae Choi, Yeseul Choi, Yujin Lee, Seung-Heon Lee, Seun Kang, Kyung-Tae Lee.

**Project administration:** Yong-Sun Bahn.

**Resources:** Kyung-Tae Lee, Yong-Sun Bahn.

**Supervision:** Yong-Sun Bahn.

**Validation:** Jin-Tae Choi.

**Visualization:** Jin-Tae Choi.

**Writing – original draft:** Jin-Tae Choi, Yong-Sun Bahn.

**Writing – review & editing:** Kyung-Tae Lee, Yong-Sun Bahn.

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
