## [Decision Letter · Decision Letter 0]

13 Jun 2023

Dear Dr. Bahn,

Thank you very much for submitting your manuscript "The hybrid RAVE complex plays V-ATPase-dependent and -independent pathobiological roles in Cryptococcus neoformans" for consideration at PLOS Pathogens. As with all papers reviewed by the journal, your manuscript was reviewed by members of the editorial board and by two independent reviewers. In light of the reviews (below this email), we would like to invite the resubmission of a significantly-revised version that takes into account the reviewers' comments.

As you will see, the reviewers were very positive about the comprehensive dataset and investigations that you have undertaken.  However, they have proposed several experiments that would strengthen the story that you present and will hopefully prove relatively straightforward to undertake.  

We cannot make any decision about publication until we have seen the revised manuscript and your response to the reviewers' comments. Your revised manuscript is also likely to be sent to reviewers for further evaluation.

Sincerely,

Robin C. May

Academic Editor

PLOS Pathogens

Alex Andrianopoulos

Section Editor

PLOS Pathogens

Kasturi Haldar

Editor-in-Chief

PLOS Pathogens

orcid.org/0000-0001-5065-158X

Michael Malim

Editor-in-Chief

PLOS Pathogens

orcid.org/0000-0002-7699-2064

Reviewer's Responses to Questions

**Part I - Summary**

Reviewer #1: This study describes in considerable detail the role of the RAVE complex in Cryptococcus. The work is novel as this has not been previously examined in Cryptococcus or other fungal pathogens. All experiments were well thought out and carried out with a high level of competence, with results presented in clear figures that were supported by the text in a way that made them easy to understand. While the study is mostly fundamental science it also has implications for drug development, as the RAVE complex contains a divergent Rav1 subunit that could be a target for antifungal agents.

While the paper is well written there are some minor points of English that if addressed would help make it a bit easier to follow in places. I have noted these on the marked-up copy of the manuscript.

Reviewer #2: This study by Choi et al. exams the role of RAVE complex components, Rav1 and Wdr1, on Cryptococcus neoformans growth, response to stress, expression of virulence factors, and virulence in animal infection models. The study identifies RAVE complex components using gene orthology to RAVE components in other eukaryotes. They identify RAV1, encoding a protein with predicted structural similarity to the S. cerevisiae Rav1 RAVE component (though phylogenetically more closely related to the human DMXL2 and DMXL1 proteins). They could not find an ortholog for S. cerevisiae Rav2 but instead an ortholog of human WDR7 RAVE component that they name WDR1. The study then investigates the function of CnRAV1 and CnWDR1 through genetic deletion and phenotypic analysis. They also identify CnSKP1, which is a predicted ortholog for S. cerevisiae SKP1, encoding a RAVE complex component that is essential. The study ultimately focuses on CnRAV1 as having the most significant impact on C. neoformans growth. The rav1∆ mutant is significantly more sensitive than wild type to a several cellular stresses, has a complete bilateral mating defect and significant unilateral mating defect, defect in cytokinesis, defect in melanin production, and is significantly less virulent in both wax moth and murine infection models. For Rav1 specifically, the authors investigate some of the mechanisms behind these mutant phenotypes, showing that endocytic trafficking is disrupted. The authors also provide evidence of V-type ATPase dependent and independent roles of Rav1 using a V-type ATPase inhibitor and acidic pH rescue experiments.

Overall, this is a comprehensive investigation of the consequences of C. neoformans RAV1 deletion. The experiments are well-done and clearly presented. The strengths of this study include the thorough investigation of rav1∆/∆ mutant phenotypes in vitro and in vivo, mechanistic investigation of endocytic trafficking using STE6-GFP localization, and biochemical evidence that Rav1 directly interacts with the other RAVE pathway component, Wdr1, and a subunit of the V-ATPase. This study demonstrates an important role for RAVE, Rav1, and the V-type ATPase in several C. neoformans processes critical for virulence and would be of significant interest to the Cryptococcus field. My main issue is that the study focuses largely on descriptive data and the study would benefit from more rigorous evidence that Rav1 and Wdr1 (and that Skp1 does not) function together through V-type ATPase.

**Part II – Major Issues: Key Experiments Required for Acceptance**

Reviewer #1: There are no major issues that need to be addressed.

Reviewer #2: My major concerns to be addressed:

1) Experimental evidence linking rav1∆ phenotypes to V-type ATPase would strengthen this study. How does bafilomycin A1 treatment impact rav1∆ phenotypes? For those that are V-type ATPase dependent, you would expect to see no difference in mutant phenotype.

2) There is no investigation of the role of SKP1 in the C. neoformans RAVE complex. The study relies primarily on data from the humans and S. cerevisiae to argue it is unlikely to be involved. However, additional evidence is necessary to determine whether it is truly uninvolved in C. neoformans. Also, the growth defect after SKP1 gene repression is modest – have the authors tried to knock out this gene in C. neoformans?

3) Expanding the analysis for Wdr1 in RAVE-dependent processes. Are the V-type ATPase-independent phenotypes dependent on Wdr1? Or does Rav1 function independent of other RAVE complex components? Are the wdr1∆ phenotypes also rescued by acidic pH?

**Part III – Minor Issues: Editorial and Data Presentation Modifications**

Reviewer #1: Minor issues are reported on the marked up copy of the manuscript.

Reviewer #2: Minor concerns:

1) The authors argue that the C. neoformans RAVE complex is a hybrid of yeast and higher eukaryotes, but the CnRAV1 ortholog is phylogenetically more closely related to human orthologs than S. cerevisiae RAV1. This could suggest that it is not a hybrid, but simply more similar to the RAVE in higher eukaryotes. More discussion and perhaps more detailed comparisons between human DXML1/2 and ScRAV1, would be helpful.

2) Given the striking growth defects of the rav1∆ mutant in vitro, it is a surprising that this strain is virulent at all in the murine infection model. Does rav1∆ acquire mutations or adapt in some way that ultimately allow for proliferation and/or dissemination within these mice?

3) Rav1∆ defects in the BBB transcytosis assay may be due primarily to growth defects at 37˚C, rather than a specific defect in transcytosis. Instead of transcytosis to cfu’s observed with WT C. neoformans, it would be better to normalize transcytosed Cn to non-transcytosed Cn for each strain.

PLOS authors have the option to publish the peer review history of their article (what does this mean?). If published, this will include your full peer review and any attached files.

Reviewer #1: No

Reviewer #2: No
---

## [Decision Letter · Decision Letter 1]

29 Sep 2023

Dear Dr. Bahn,

We are pleased to inform you that your manuscript 'The hybrid RAVE complex plays V-ATPase-dependent and -independent pathobiological roles in Cryptococcus neoformans' has been provisionally accepted for publication in PLOS Pathogens.

Best regards,

Robin C. May

Academic Editor

PLOS Pathogens

Alex Andrianopoulos

Section Editor

PLOS Pathogens

Kasturi Haldar

Editor-in-Chief

PLOS Pathogens

orcid.org/0000-0001-5065-158X

Michael Malim

Editor-in-Chief

PLOS Pathogens

orcid.org/0000-0002-7699-2064

Reviewer Comments (if any, and for reference):

Reviewer's Responses to Questions

**Part I - Summary**

Reviewer #2: This study by Choi et al. exams the role of RAVE complex components, Rav1 and Wdr1, on Cryptococcus neoformans growth, response to stress, expression of virulence factors, and virulence in animal infection models. The study identifies RAVE complex components using gene orthology to RAVE components in other eukaryotes. They identify RAV1, encoding a protein with predicted structural similarity to the S. cerevisiae Rav1 RAVE component (though phylogenetically more closely related to the human DMXL2 and DMXL1 proteins). They could not find an ortholog for S. cerevisiae Rav2 but instead an ortholog of human WDR7 RAVE component that they name WDR1. The study then investigates the function of CnRAV1 and CnWDR1 through genetic deletion and phenotypic analysis. They also identify CnSKP1, which is a predicted ortholog for S. cerevisiae SKP1, encoding a RAVE complex component. The authors demonstrate that CnSKP1 is essential and based on protein interactions studies, unlikely to be a component of the RAVE complex. The study ultimately focuses on CnRAV1 as having the most significant impact on C. neoformans growth. The rav1∆ mutant is significantly more sensitive than wild type to a several cellular stresses, has a complete bilateral mating defect and significant unilateral mating defect, defect in cytokinesis, defect in melanin production, and is significantly less virulent in both wax moth and murine infection models. For Rav1 specifically, the authors investigate some of the mechanisms behind these mutant phenotypes, showing that endocytic trafficking is disrupted. The authors also provide evidence of V-type ATPase dependent and independent roles of Rav1 using a V-type ATPase inhibitor and acidic pH rescue experiments.

This is a well-presented and thorough study that demonstrates that Rav1 and the V-type ATPase is required for C. neoformans virulence. The authors addressed all of my major concerns completely. This study will be of significant interest to the Cryptococcus field.

**Part II – Major Issues: Key Experiments Required for Acceptance**

Reviewer #2: none

**Part III – Minor Issues: Editorial and Data Presentation Modifications**

Reviewer #2: none

PLOS authors have the option to publish the peer review history of their article (what does this mean?). If published, this will include your full peer review and any attached files.

Reviewer #2: No

---

## [Editor Report · Acceptance letter]

4 Oct 2023

Dear Dr. Bahn,

We are delighted to inform you that your manuscript, "The hybrid RAVE complex plays V-ATPase-dependent and -independent pathobiological roles in Cryptococcus neoformans," has been formally accepted for publication in PLOS Pathogens.

Best regards,

Kasturi Haldar

Editor-in-Chief

PLOS Pathogens

orcid.org/0000-0001-5065-158X

Michael Malim

Editor-in-Chief

PLOS Pathogens

orcid.org/0000-0002-7699-2064